# EQUIVARIANT ENTITY-RELATIONSHIP NETWORKS

## ABSTRACT

The relational model is a ubiquitous representation of big-data, in part due to its extensive use in databases. However, recent progress in deep learning with relational data has been focused on (knowledge) graphs. In this paper we propose Equivariant Entity-Relationship Networks, a general class of parameter-sharing neural networks derived from the entity-relationship model. We prove that our proposed feed-forward layer is the most expressive linear layer under the given equivariance constraints, and subsumes recently introduced equivariant models for sets, exchangeable tensors, and graphs. The proposed feed-forward layer has linear complexity in the data and can be used for both inductive and transductive reasoning about relational databases, including database embedding, and the prediction of missing records. This provides a principled theoretical foundation for the application of deep learning to one of the most abundant forms of data.

## 1 INTRODUCTION

In the relational model of data, we have a set of *entities*, and one or more *instances* of each entity. These instances interact with each other through a set of fixed *relations* between entities. A set of *attributes* may be associated with each type of entity and relation.[1] This simple idea is widely used to represent data, often in the form of a relational database, across a variety of domains, from shopping records, social networking data, and health records, to heterogeneous data from astronomical surveys.

Learning and inference on relational data has been the topic of research in machine learning over the past decades. The relational model is closely related to first order and predicate logic, where the existence of a relation between instances becomes a truth statement about a world. The same formalism is used in AI through a probabilistic approach to logic, where the field of *statistical relational learning* has fused the relational model with the framework of *probabilistic graphical models*. Examples of such models include plate models, probabilistic relational models, Markov logic networks, and relational dependency networks (Getoor & Taskar, 2007).

A closely related area that has enjoyed accelerated growth in recent years is *relational and geometric deep learning*, where the term "relational" is used to denote the inductive bias introduced by a graph structure. Although relational and graph-based terms are often used interchangeably in the machine learning community, they could refer to different data structures: graph-based models such as graph databases (Robinson et al., 2013) and knowledge graphs simply represent data as an attributed (hyper-)graph, while the relational model, common in databases, uses the entity-relation (ER) diagram (Chen, 1976) to constrain the relations of each instance (corresponding to a node), based on its entity-type; see Fig. 1(a).

Inspired by the success of equivariant deep learning, we use *invariance and equivariance* to encode the structure of relational data. This type of inductive bias informs a model's behaviour under various transformations. Equivariant models have been successfully used for deep learning on data with a variety of structures, from translation equivariant images (LeCun et al., 1998), to geometric settings (Cohen et al., 2018a; 2019), to discrete objects such as sets (Zaheer et al., 2017), and graphs (Kondor et al., 2018b). By adopting this perspective, we present a maximally expressive feed-forward layer that achieves equivariance w.r.t. exchangeabilities in relational data as it is expressed by the entity-relationship model. Our feedforward layer generalizes recently proposed layers for sets (Zaheer et al., 2017), exchangeable tensors (Hartford et al., 2018), and graphs (Maron et al., 2018).

---

[1]An alternative terminology refers to instances and entities as entities and entity types.

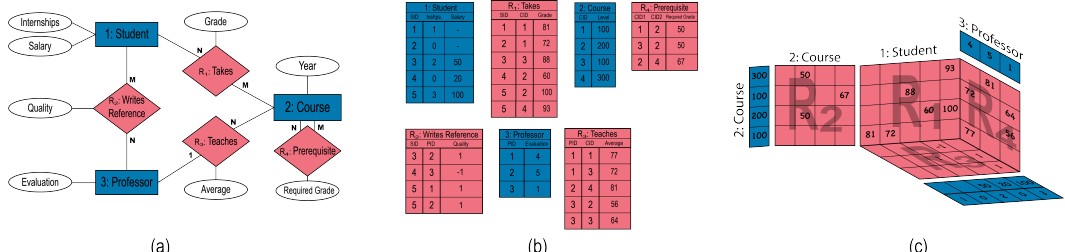

Figure 1: (**a**) The Entity-Relationship (ER) diagram for our running example. There are three entities: STUDENT, COURSE and PROFESSOR (labeled 1,2 and 3 respectively), and four pairwise relations: TAKES (STUDENT-COURSE, represented by $\mathbb{R}_1 = \{1, 2\}$); the self-self relation PREREQUISITE (COURSE-COURSE, represented by $\mathbb{R}_2 = \{2, 2\}$); WRITES REFERENCE (STUDENT-PROFESSOR, represented by $\mathbb{R}_3 = \{1, 3\}$); and TEACHES (PROFESSOR-COURSE, represented by $\mathbb{R}_4 = \{2, 3\}$). The full set of relations is $\mathfrak{R} = \{\{1\}, \{2\}, \{3\}, \{1, 2\}, \{1, 3\}, \{2, 3\}, \{2, 2\}\}$. Both entities and relations have associated attributes — *e.g.*, when a STUDENT takes a COURSE, they receive a GRADE. The singleton relation $\mathbb{R} = \{1\}$ can encode STUDENT attribute(s) such as number of INTERNSHIPS, or SALARY after graduating. Since each COURSE has a single PROFESSOR as its teacher, this relation is one-to-many. (**b**) Some possible relational tables in an instantiation of the ER diagram of (a). There are $N_1 = 5$ instances of STUDENT, $N_2 = 4$ instances of COURSE and $N_3 = 3$ instances of PROFESSOR. The attributes associated with each entity and relation are stored in the corresponding table —*e.g.*, table $\mathbf{X}^{\{1,2\}}$ suggests that the STUDENT 5 took COURSE 4 and received a GRADE of 93. (**c**) Sparse tensor representation $\mathbf{X}^{\mathbb{R}_1}, \ldots, \mathbf{X}^{\mathbb{R}_7}$ of the tables of (b). The vectorized form of this set of sparse tensors, $\mathrm{vec}(\mathbb{X}) = \langle \mathrm{vec}(\mathbf{X}^{\{1\}}); \ldots; \mathrm{vec}(\mathbf{X}^{\{2,2\}}) \rangle$ —the column-vector concatenation of $\mathrm{vec}(\mathbf{X}^{\mathbb{R}})$'s— is the input to our feed-forward layer $\sigma(\mathbf{W} \, \mathrm{vec}(\mathbb{X}))$. Here, the parameter-tying in the weight matrix $\mathbf{W}$ of the EERL (Fig. 2) guarantees that any permutation of elements of $\mathrm{vec}(\mathbb{X})$ corresponding to a shuffling of entities in this tensor representation (and only these permutations), results in the same permutation of the output of the layer.

## 2 REPRESENTATIONS OF RELATIONAL DATA

We represent the relational model by a set of *entities* $\mathcal{D} = \{1, ..., D\}$ and a set of *relations* $\mathfrak{R} \subseteq 2^{\mathcal{D}}$, indicating how the entities interact with one another. For each entity $d \in \mathcal{D}$ we have a set $\{1, .., N_d\}$ of *instances* of that entity. For each relation $\mathbb{R} \in \mathfrak{R}$, where $\mathbb{R} = \{d_1, \ldots, d_{|\mathbb{R}|}\}$ is a set of entities, we observe data in the form of a set of tuples $\mathbb{X}_{\mathbb{R}} = \{\langle n_{d_1}, \ldots, n_{d_{|\mathbb{R}|}}, x \rangle \mid n_{d_i} \in [N_{d_i}], x \in \mathbb{R}^K\}$. That is, for each element of $\mathbb{X}_{\mathbb{R}}$ we associate the feature vector $x$ with the relationship between the instances indexed by $n_{d_i}$ for each entity $d_i \in \mathbb{R}$. See Fig. 1 for a detailed, concrete example. Note that the *singleton relation* $\mathbb{R} = \{d_i\}$ can be used to incorporate individual entity attributes (such as professors' evaluations in Fig. 1(a)).

In the most general case, we allow for both $\mathfrak{R}$, and any $\mathbb{R} \in \mathfrak{R}$ to be multisets (*i.e.*, to contain duplicate entries). $\mathfrak{R}$ is a multiset if we have multiple relations between the same set of entities. For example, we may have a SUPERVISES relation between STUDENTS and PROFESSORS, in addition to the WRITES REFERENCE relation. A particular relation $\mathbb{R}$ is a multiset if it contains multiple copies of the same entity. Such relations are ubiquitous in the real world, describing for example, the connection graph of a social network, the sale/purchase relationships between a group of companies, or, in our running example, the COURSE-COURSE relation capturing prerequisite information. For our derivations we make the simplifying assumption that each *attribute* $x \in \mathbb{R}$ is a scalar. Extension to $\mathbf{x} \in \mathbb{R}^K$ using multiple channels is trivial and discussed in Appendix F. Another common feature of relational data, the one-to-many relation, is addressed in Appendix E.

### 2.1 TUPLES, TABLES AND TENSORS

In relational databases the set of tuples $\mathbb{X}^{\mathbb{R}}$ is often represented using a table, with one row for each tuple; see Fig. 1(b). An equivalent representation for $\mathbb{X}^{\mathbb{R}}$ is using a "sparse" $|\mathbb{R}|$-dimensional tensor $\mathbf{X}^{\mathbb{R}} \in \mathbb{R}^{N_{d_1} \times \ldots \times N_{d_{|\mathbb{R}|}}}$, where each dimension of this tensor corresponds to an entity $d \in \mathbb{R}$, and the the length of that dimension is the number of instances $N_d$. In other words

$$\langle n_{d_1}, \ldots, n_{d_{|\mathbb{R}|}}, x \rangle \in \mathbb{X}^{\mathbb{R}} \quad \Leftrightarrow \quad \mathbf{X}^{\mathbb{R}}_{n_{d_1}, \ldots, n_{d_{|\mathbb{R}|}}} = x.$$

We work with this tensor representation of relational data. We use $\mathbb{X} = \{\mathbf{X}^{\mathbb{R}} \mid \mathbb{R} \in \mathfrak{R}\}$ to denote the set of all sparse tensors that define the relational data(base); see Fig. 1(c). For the following

discussions around exchangeability and equivariance, we assume that for all $\mathbb{R}$, $\mathbf{X}^{\mathbb{R}}$ are fully observed, dense tensors. Subsequently, we will discard this assumption and attempt to make predictions for (any subset of) the missing records.

Note that relations $\mathbb{R}$ can be multisets. For simplicity, we handle this in the main text by considering equal elements as distinct through indexing (*e.g.*, $d^{(i)} = d^{(j)}$), while leaving a formal definition of multisets for the supplementary material; see Appendix B.

## 3 EXCHANGEABILITIES OF ENTITY-RELATIONSHIP DATA

Recall that in the representation $\mathbb{X}$, each entity $d \in \{1, ..., D\}$ has a set of *instances* indexed by $n_d \in \{1, .., N_d\}$. The ordering of $\{1, ..., N_d\}$ is arbitrary, and we can shuffle these instances, affecting only the representation, and not the "content" of the relational data. However, in order to maintain consistency across data tables, we also have to shuffle all the tensors $\mathbf{X}^{\mathbb{R}}$, where $d \in \mathbb{R}$, using the same permutation applied to the tensor dimension corresponding to $d$. At a high level, this simple indifference to shuffling defines the exchangeabilities of relational data. A mathematical group formalizes this idea.

A mathematical *group* is a set equipped with a binary operation between its members, such that the set and the operation satisfy closure, associativity, invertability and existence of a unique identity element. $\mathcal{S}^M$ refers to the *symmetric group*, the group of all permutations of $M$ objects. A natural *representation* for a member of this group $g^M \in \mathcal{S}^M$, is a permutation matrix $\mathbf{G} \in \{0, 1\}^{M \times M}$. Here, the binary group operation is the same as the product of permutation matrices. In this notation, $\mathcal{S}^{N_d}$ is the group of all permutations of instances $1, \ldots N_d$ of entity $d \in \{1, .., D\}$. To consider permutations to multiple dimensions of a data tensor we can use the direct product of groups. Given two groups $\mathcal{G}$ and $\mathcal{H}$, the direct product $\mathcal{G} \times \mathcal{H}$ is defined by

$$(g, \hbar) \in \mathcal{G} \times \mathcal{H} \Leftrightarrow g \in \mathcal{G}, \hbar \in \mathcal{H} \quad \text{and} \quad (g, \hbar) \circ (g', \hbar') = (g \circ g', \hbar \circ \hbar'). \tag{1}$$

That is, the underlying set is the Cartesian product of the underlying sets of $\mathcal{G}$ and $\mathcal{H}$, and the group operation is the component-wise operation.

Observe that we can associate the group $\mathcal{S}^{N_1} \times \ldots \times \mathcal{S}^{N_D}$ with the $D$ entities in a relational model, where each entity $d$ has $N_d$ instances. Intuitively, applying permutations from this group to the corresponding relational data should not affect the underlying contents, while applying permutations from outside this group should. To see this, consider the tensor representation of Fig. 1(c): permuting students, courses or professors shuffles rows or columns of $\mathbb{X}$, but preserves its underlying content. However, arbitrary shuffling of the elements of this tensor could alter its content.

Our goal is to define a feed-forward layer that is "aware" of this structure. For this, we first need to formalize the *action* of $\mathcal{S}^{N_1} \times \ldots \times \mathcal{S}^{N_D}$ on the vectorized form of $\mathbb{X}$.

**Vectorization.** For each tensor $\mathbf{X}^{\mathbb{R}} \in \mathbb{X}$, $N_{\mathbb{R}} = \prod_{d \in \mathbb{R}} N_d$ refers to the total *number of elements* of tensor $\mathbf{X}^{\mathbb{R}}$ (note that for now we are assuming that the tensors are dense). We will refer to $N = \sum_{\mathbb{R} \in \mathfrak{R}} N_{\mathbb{R}}$ as the number of elements of $\mathbb{X}$. Then $\text{vec}(\mathbf{X}^{\mathbb{R}}) \in \mathbb{R}^{N_{\mathbb{R}}}$ refers to the *vectorization* of $\mathbf{X}^{\mathbb{R}}$, obtained by successively stacking its elements along its dimensions, where the order of dimensions is given by $d \in \mathbb{R}$. We use $\text{vec}^{-1}(\cdot)$ to refer to the inverse operation of $\text{vec}(\cdot)$, so that $\text{vec}^{-1}(\text{vec}(\mathbf{X})) = \mathbf{X}$. With a slight abuse of notation, we use $\text{vec}(\mathbb{X}) \in \mathbb{R}^N$ to refer to $[\text{vec}(\mathbf{X}^{\mathbb{R}_1}); \ldots; \text{vec}(\mathbf{X}^{\mathbb{R}_{|\mathfrak{R}|}})]$, the vectorized form of the entire relational data. The weight matrix $\mathbf{W}$ that we define later creates a feed-forward layer $\sigma(\mathbf{W} \, \text{vec}(\mathbb{X}))$ applied to this vector.

**Group Action.** The *action* of $g \in \mathcal{S}^{N_1} \times \ldots \times \mathcal{S}^{N_D}$ on $\text{vec}(\mathbb{X}) \in \mathbb{R}^N$, permutes the elements of $\text{vec}(\mathbb{X})$. Our objective is to define this group action by mapping $\mathcal{S}^{N_1} \times \ldots \times \mathcal{S}^{N_D}$ to a group of permutations of all $N = \sum_{\mathbb{R} \in \mathfrak{R}} \prod_{d \in \mathbb{R}} N_d$ entries of the database – *i.e.*, a homomorphism into $\mathcal{S}^N$. To this end we need to use two types of matrix product. Let $\mathbf{G} \in \mathbb{R}^{N_1 \times N_2}$ and $\mathbf{H} \in \mathbb{R}^{N_3 \times N_4}$ be two matrices. The *direct sum* $\mathbf{G} \oplus \mathbf{H}$ is an $(N_1 + N_3) \times (N_2 + N_4)$ block-diagonal matrix, and the

*Kronecker product* $\mathbf{G} \otimes \mathbf{H}$ is an $(N_1 N_3) \times (N_2 N_4)$ matrix:

$$\mathbf{G} \oplus \mathbf{H} \doteq \begin{pmatrix} \mathbf{G} & \mathbf{0} \\ \mathbf{0} & \mathbf{H} \end{pmatrix}, \quad \mathbf{G} \otimes \mathbf{H} \doteq \begin{pmatrix} \mathbf{G}_{1,1}\mathbf{H} & \dots & \mathbf{G}_{1,N_2}\mathbf{H} \\ \vdots & \ddots & \vdots \\ \mathbf{G}_{N_1,1}\mathbf{H} & \dots & \mathbf{G}_{N_1,N_2}\mathbf{H} \end{pmatrix}.$$

Note that in the special case that both $\mathbf{G}$ and $\mathbf{H}$ are permutation matrices, $\mathbf{G} \oplus \mathbf{H}$ and $\mathbf{G} \otimes \mathbf{H}$ will also be permutation matrices. Both of these matrix operations can represent the direct product of permutation groups. That is, given two permutation matrices $\mathbf{G}^1 \in \mathcal{S}^{N_1}$, and $\mathbf{G}^2 \in \mathcal{S}^{N_2}$, we can use both $\mathbf{G}^1 \otimes \mathbf{G}^2$ and $\mathbf{G}^1 \oplus \mathbf{G}^2$ to represent members of $\mathcal{S}^{N_1} \times \mathcal{S}^{N_2}$. However, the resulting permutation matrices, can be interpreted as different *actions*: while the $(N_1 + N_2) \times (N_1 + N_2)$ direct sum matrix $\mathbf{G}^1 \oplus \mathbf{G}^2$ is a permutation of $N_1+N_2$ objects, the $(N_1 N_2) \times (N_1 N_2)$ Kronecker product matrix $\mathbf{G}^1 \otimes \mathbf{G}^2$ is a permutation of $N_1 N_2$ objects.

---

**Claim 1.** *Consider the vectorized relational data* $\mathrm{vec}(\mathbb{X})$ *of length* $N$. *The action of* $\mathcal{S}^{N_1} \times \dots \times \mathcal{S}^{N_D}$ *on* $\mathrm{vec}(\mathbb{X})$ *is given by the following permutation group*

$$\mathcal{G}^{\mathbb{X}} \doteq \big\{ \bigoplus_{\mathbb{R} \in \mathfrak{R}} \bigotimes_{d \in \mathbb{R}} \mathbf{G}^d \mid (\mathbf{G}^1, \dots, \mathbf{G}^D) \in \mathcal{S}^{N_1} \times \dots \times \mathcal{S}^{N_D} \big\}. \tag{2}$$

*where the order of relations in* $\bigoplus_{\mathbb{R} \in \mathfrak{R}}$ *is consistent with the ordering used for vectorization of* $\mathbb{X}$.

---

*Proof.* The Kronecker product $\bigotimes_{d \in \mathbb{R}} \mathbf{G}^d$ when applied to $\mathrm{vec}(\mathbf{X}^{\mathbb{R}})$, permutes the underlying tensor $\mathbf{X}^{\mathbb{R}}$ along the axes $d \in \mathbb{R}$. Using direct sum, these permutations are applied to each tensor $\mathrm{vec}(\mathbf{X}^{\mathbb{R}})$ in $\mathrm{vec}(\mathbb{X}) = [\mathrm{vec}(\mathbf{X}^{\mathbb{R}_1}); \dots, \mathrm{vec}(\mathbf{X}^{\mathbb{R}_D})]$. The only constraint, enforced by (2) is to use the same permutation matrix $\mathbf{G}^d$ for all $\mathbb{R}$ when $d \in \mathbb{R}$. Therefore any matrix-vector product $\mathbf{G}^N \mathrm{vec}(\mathbb{X})$ is a "legal" permutation of $\mathrm{vec}(\mathbb{X})$, since it only shuffles the instances of each entity. $\square$

## 4 EQUIVARIANT ENTITY-RELATIONSHIP LAYER

Our objective is to constrain the standard feed-forward layer $f : \mathbb{R}^{K \times N} \to \mathbb{R}^{K' \times N}$ —where $K, K'$ are the number of input and output channels, and $N = |\mathrm{vec}(\mathbb{X})|$— such that any "legal" transformation of the input, as defined in (2), should result in the same transformation of the output. Doing so amounts to augmenting our input data by applying all such transformations and using an unconstrained feed-forward layer. This layer would be quadratic in the size of the database, and the model would require exponentially more data to train. In the case of even moderately sized databases this approach is simply infeasible, further motivation the our EERL (Definition 1). For clarity, we limit the following definition to the case where $K = K' = 1$; see Appendix F for the case of multiple channels.

---

**Definition 1** (Equivariant Entity-Relationship Layer; EERL)**.** *Let* $\mathbf{G}^{\mathbb{X}}$ *be any* $N \times N$ *permutation of* $\mathrm{vec}(\mathbb{X})$. *A fully connected layer* $\sigma(\mathbf{W} \mathrm{vec}(\mathbb{X}))$ *with* $\mathbf{W} \in \mathbb{R}^{N \times N}$ *is called an Equivariant Entity-Relationship Layer if*

$$\sigma(\mathbf{W}\mathbf{G}^{\mathbb{X}} \mathrm{vec}(\mathbb{X})) = \mathbf{G}^{\mathbb{X}}\sigma(\mathbf{W} \mathrm{vec}(\mathbb{X})) \quad \forall \mathbb{X} \quad \Leftrightarrow \quad \mathbf{G}^{\mathbb{X}} \in \mathcal{G}^{\mathbb{X}}. \tag{3}$$

*That is, an EERL is a layer that commutes with the permutation* $\mathbf{G}^{\mathbb{X}}$ *if and only if* $\mathbf{G}^{\mathbb{X}}$ *is a legal permutation, as defined by (2).*

---

An EERL[2] captures the notion that the order of the entities in the relational data $\mathbb{X}$ is not important, and the layer should not produce different outputs when given different orderings. We now propose a procedure to tie the entries of $\mathbf{W}$ so as to guarantee the conditions of Definition 1. Moreover, we show that the resulting linear map $\mathbf{W} : \mathbb{R}^{N \times N}$ is the most expressive linear map under the equivariance constraints.

---

[2]In its most general form, we can also allow for particular kinds of *bias* parameters in Definition 1; see Appendix D for parameter-tying in the bias term.

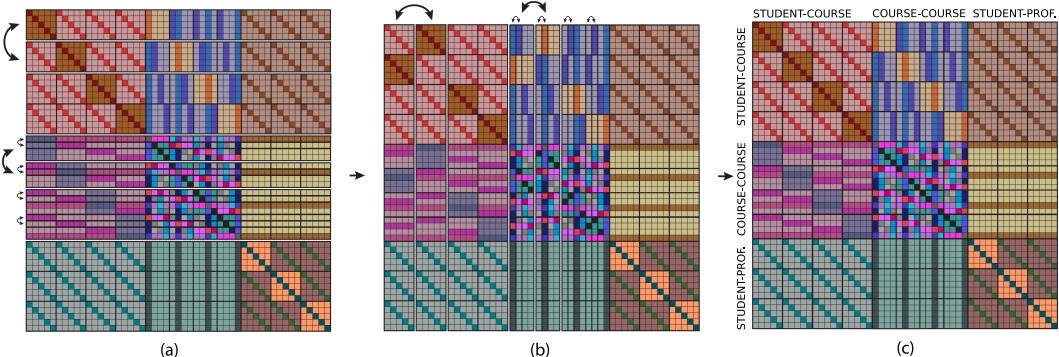

Figure 2: (**a**) The parameter matrix $\mathbf{W}$ from Example 2. Each colour represents a unique parameter value. The nine blocks, showing the interaction of three relations, are clearly visible. The arrows indicate the permutation that is being applied. (b) The result of applying a permutation to the rows of $\mathbf{W}$. The permutation is $\mathbf{G} = \mathbf{G}^1 \otimes \mathbf{G}^2 \otimes \mathbf{G}^3$ permuting the instances of STUDENT, COURSE and PROFESSOR. Here, $\mathbf{G}^1 \in \mathcal{S}^5$ and $\mathbf{G}^3 \in \mathcal{S}^3$ are simply the respective identity permutations, while $\mathbf{G}^2 \in \mathcal{S}^4$ is the permutation $(2, 1, 3, 4)$ (*i.e.*, swap the first and second items). This corresponds to a "legal" permutation as defined by (2). We can see that this swapping is applied block-wise to blocks of rows of $\mathbf{W}$ corresponding to each $\mathbf{W}^{i,j}$ for which $2 \in \mathbb{R}_i$. In the case of $\mathbb{R}_2 = \{2, 2\}$, $\mathbf{G}^2$ must also be applied to rows within blocks, as these correspond to entity 2 as well. (c) The result of applying the inverse permutation (in this case, the same permutation) to the columns of $\mathbf{W}$. Here, we swap columns block-wise in each $\mathbf{W}^{i,j}$ for which $2 \in \mathbb{R}_j$. By doing so we recover the original matrix. This symmetry condition on $\mathbf{W}$ is essential in achieving equivariance (see Lemma 1 in the proof of Theorem 4.1).

We build up the matrix $\mathbf{W} \in \mathbb{R}^{N \times N}$ block-wise, with blocks $\mathbf{W}^{i,j} \in \mathbb{R}^{N_{\mathbb{R}_i} \times N_{\mathbb{R}_j}}$ corresponding to each pair of relations $\mathbb{R}_i, \mathbb{R}_j$:

$$
\mathbf{W} = \begin{pmatrix} \mathbf{W}^{1,1} & \mathbf{W}^{1,2} & \dots & \mathbf{W}^{1,|\mathfrak{R}|} \\ \vdots & \vdots & \ddots & \vdots \\ \mathbf{W}^{|\mathfrak{R}|,1} & \mathbf{W}^{|\mathfrak{R}|,2} & \dots & \mathbf{W}^{|\mathfrak{R}|,|\mathfrak{R}|} \end{pmatrix}. \tag{4}
$$

We tie parameters within each block, and not across blocks. To concisely express the somewhat complex tying scheme, we use the following indexing notation.

**Indexing Notation.** The parameter block $\mathbf{W}^{i,j}$ is an $N_{\mathbb{R}_i} \times N_{\mathbb{R}_j}$ matrix, where $N_{\mathbb{R}_i} = \prod_{d \in \mathbb{R}_i} N_d$. We want to index rows and columns of $\mathbf{W}^{i,j}$. Given the relation $\mathbb{R}_i = \{d_1, \dots, d_{|\mathbb{R}_i|}\}$, we use the tuple $\mathbf{n}^i \doteq \langle n^i_{d_1}, \dots, n^i_{d_{|\mathbb{R}_i|}} \rangle \in [N_{d_1}] \times \dots \times [N_{d_{|\mathbb{R}_i|}}]$ to index an element in the set $[N_{\mathbb{R}_i}]$. Since each element of $\mathbf{n}^i$ indexes instances of a particular entity, $\mathbf{n}^i$ can be used as an index both for data block $\mathrm{vec}(\mathbf{X}^{\mathbb{R}_i})$ and for the rows of parameter block $\mathbf{W}^{i,j}$. In particular, to denote an entry of $\mathbf{W}^{i,j}$, we use $\mathbf{W}^{i,j}_{\mathbf{n}^i, \mathbf{n}^j}$. Moreover, we use $n^i_d$ to denote the element of $\mathbf{n}^i$ corresponding to entity $d \in \mathbb{R}_i$. Note that this is not necessarily the $d^{th}$ element of the tuple $\mathbf{n}^i$. For example, if $\mathbb{R}_i = \{1, 4, 5\}$ and $\mathbf{n}^i = \langle 400, 12, 3 \rangle$, then $n^i_4 = 12$ and $n^i_5 = 3$. When $\mathbb{R}_i$ is a multiset, we can use $n^i_{d^{(k)}}$ to refer the to the element of $\mathbf{n}^i$ corresponding to the $k$-th occurrence of entity $d$ (where the order corresponds to the ordering of elements in $\mathbf{n}^i$). Table 1 in the Appendix summarizes our notation.

## 4.1 PARAMETER TYING

Let $\mathbf{W}^{i,j}_{\mathbf{n}^i, \mathbf{n}^j}$ and $\mathbf{W}^{i,j}_{\mathbf{m}^i, \mathbf{m}^j}$ denote two arbitrary elements of the parameter matrix $\mathbf{W}^{i,j}$. Our objective is to decide whether or not they should be tied together to ensure the resulting layer is an EERL (Definition 1). To this end we define an equivalence relation between the indices: $\langle \mathbf{n}^i, \mathbf{n}^j \rangle \sim \langle \mathbf{m}^i, \mathbf{m}^j \rangle$, and tie together all entries of $\mathbf{W}$ that are equivalent under this relation. Index $\langle \mathbf{n}^i, \mathbf{n}^j \rangle$ is equivalent to index $\langle \mathbf{m}^i, \mathbf{m}^j \rangle$ if they have the same equality patterns over their indices for each unique entity $d$.

We consider $\mathbf{n}^{i,j}$, the concatenation of $\mathbf{n}^i$ with $\mathbf{n}^j$, and examine the sub-tuples $\mathbf{n}^{i,j}_d$, where $\mathbf{n}^{i,j}$ is restricted to only those indices that index entity $d$. We do this so that we can ask if these indices refer

to the same instance (*e.g.*, the same student), and we can only meaningfully compare indices of the same entity (*e.g.*, two entries in the STUDENT table may refer to the same student, but an entry in the STUDENT table cannot refer to the same thing as an entry in the COURSE table. Also, recall that we allow $\mathbb{R}_i$ and $\mathbb{R}_j$ to be multisets (*e.g.*, the COURSE-COURSE relation $\mathbb{R}_4$ in Fig. 1). We say two tuples $\mathbf{n}_d^{i,j} = \langle n_{d^{(1)}}, \dots n_{d^{(L)}} \rangle$ and $\mathbf{m}_d^{i,j} = \langle m_{d^{(1)}}, \dots m_{d^{(L)}} \rangle$ are equivalent iff the have the same equality pattern $n_{d^{(l)}} = n_{d^{(l')}} \Leftrightarrow m_{d^{(l)}} = m_{d^{(l')}} \forall l \in [L]$. Accordingly, two index tuples $\mathbf{n}^{i,j}$ and $\mathbf{m}^{i,j}$ are equivalent iff they are equivalent for all $d \in \mathbb{R}_i \cup \mathbb{R}_j$.

Because of this tying scheme, the total number of *free* parameters in $\mathbf{W}^{i,j}$ is the product of the number of possible different partitionings of $\mathbf{n}_d^{i,j}$ for each unique entity $d \in \mathbb{R}_i \cup \mathbb{R}_j$, and so is a product of Bell numbers[3], which count the number of partitions of a set given size; see Appendix C for details. This relation to Bell numbers was previously shown for equivariant graph networks (Maron et al., 2018) which, as we see in Section 6, are closely related to, and indeed a special case of, our model. This parameter-sharing scheme admits a simple recursive form, if the database has no self-relations (*i.e.*, the $\mathbb{R}_i$ are not multisets); see Appendix E.

**Example 1.** *[Fig. 2(a)] To get an intuition for this tying scheme, consider a simplified version of the relational structure of Fig. 1, restricted to the three relations $\mathbb{R}_1 = \{1, 2\}$, self-relation $\mathbb{R}_2 = \{2, 2\}$, and $R_3 = \{1, 3\}$ with $N_1 = 5$ STUDENTS, $N_2 = 4$ COURSES, and $N_3 = 3$ PROFESSORS. Then $N = 5 \times 4 + 4 \times 4 + 5 \times 3 = 51$, so $\mathbf{W} \in \mathbb{R}^{51 \times 51}$ and will have nine blocks: $\mathbf{W}^{1,1} \in \mathbb{R}^{20 \times 20}$, $\mathbf{W}^{1,2} \in \mathbb{R}^{20 \times 16}$. $\mathbf{W}^{2,2} \in \mathbb{R}^{16 \times 16}$ and so on. We use tuple $\mathbf{n}^1 = \langle n_1^1, n_2^1 \rangle \in [N_1] \times [N_2] = [5] \times [4]$ to index the rows and columns of $\mathbf{W}^{1,1}$. We also use $\mathbf{n}^1$ to index the rows of $\mathbf{W}^{1,2}$, and use $\mathbf{n}^2 = \langle n_{2^{(1)}}^2, n_{2^{(2)}}^2 \rangle \in [N_2] \times [N_2] = [4] \times [4]$ to index its columns. Other blocks are indexed similarly. Suppose $\mathbf{n}^1 = \langle 1, 4 \rangle$, $\mathbf{n}^2 = \langle 4, 5 \rangle$, $\mathbf{m}^1 = \langle 2, 3 \rangle$ and $\mathbf{m}^2 = \langle 3, 2 \rangle$ and we are trying to determine whether $\mathbf{W}_{\mathbf{n}^1, \mathbf{n}^2}^{1,2}$ and $\mathbf{W}_{\mathbf{m}^1, \mathbf{m}^2}^{1,2}$ should be tied. Then $\mathbf{n}^{1,2} = \langle 1, 4, 4, 5 \rangle$ and $\mathbf{m}^{1,2} = \langle 2, 3, 3, 2 \rangle$. When we compare the sub-tuples restricted to unique entities, we see that the equality pattern of $\mathbf{n}_1^{1,2} = \langle 1 \rangle$ matches that of $\mathbf{m}_1^{1,2} = \langle 2 \rangle$ (since it is only a singleton, it matches trivially), and the equality pattern of $\mathbf{n}_2^{1,2} = \langle 4, 4, 5 \rangle$ matches that of $\mathbf{m}_2^{1,2} = \langle 3, 3, 2 \rangle$. So these weights should be tied.*

We now establish the optimality of our constrained linear layer for relational data.

**Theorem 4.1.** *Let $\mathbb{X} = \{\mathbf{X}^{\mathbb{R}} \mid \mathbb{R} \in \mathfrak{R}\}$ be the tensor representation of some relational data and $\mathrm{vec}(\mathbb{X})$ its vectorized form. If we define $\mathbf{W}$ block-wise according to (4), with blocks given by the tying scheme above, then the feed-forward layer $\sigma(\mathbf{W} \, \mathrm{vec}(\mathbb{X}))$ is an Equivariant Entity-Relationship Layer (Definition 1).*

This theorem assures us that a layer constructed with our parameter-sharing scheme achieves equivariace w.r.t. the exchangeabilities of the relational data. However, one may wonder whether an alternative, more expressive parameter-sharing scheme (with the same form of output) may be possible. The following theorem asserts that this is not the case, and that our model is the most expressive parameter-tying scheme possible for an EERL.

**Theorem 4.2.** *Let $\mathbb{X} = \{\mathbf{X}^{\mathbb{R}} \mid \mathbb{R} \in \mathfrak{R}\}$ be the tensor representation of some relational data. If $\mathbf{W} \in \mathbb{R}^{N \times N}$ is any parameter matrix other than the one defined above, then the feed-forward layer $\sigma(\mathbf{W} \, \mathrm{vec}(\mathbb{X}))$ is NOT an Equivariant Entity-Relationship Layer (Definition 1).*

In practice, we approach this constrained layer in a much more efficient way: the operations of the layer can be performed efficiently using pooling and broadcasting over tensors; see Appendix E.

## 5 EXPERIMENTS

### 5.1 SYNTHETIC DATA

To continue with our running example we synthesize a toy dataset, restricted to $\mathbb{R}_1 = \{1, 2\}$ (STUDENT-COURSE), $\mathbb{R}_2 = \{1, 3\}$ (STUDENT-PROFESSOR), and $\mathbb{R}_3 = \{2, 3\}$ (PROFESSOR-COURSE).[4] Each matrix

---

[3]The $k$-th Bell number counts the number of ways of partitioning a set of size $k$.

[4]The code for our experiments is available at <anonymous> (synthetic data) and <anonymous> (soccer data).

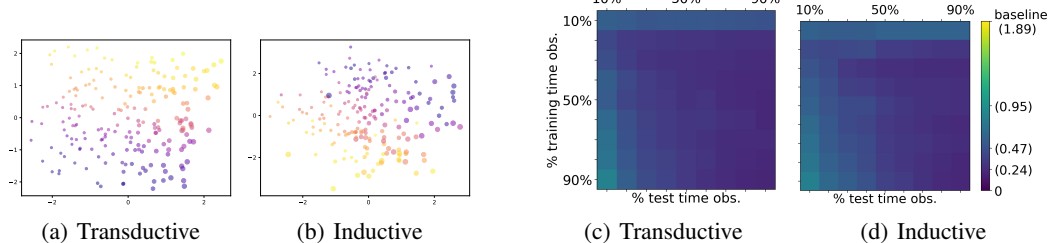

| (a) Transductive | (b) Inductive | (c) Transductive | (d) Inductive |

Figure 3: **Left (Embeddings).** Ground truth versus predicted embedding for COURSE instances in the transductive (a) and inductive (b) settings. The x-y location of each point encodes the prediction $\widehat{\mathbf{Z}^d}$, while the size and color of each encodes the ground-truth $\mathbf{Z}^d$. In the inductive setting, training and test databases contain completely distinct instances (i.e., completely different courses). **Right (Missing record prediction).** Average mean squared error in predicting missing records in STUDENT-COURSE as a function of sparsity level of the whole database $\mathbb{X}$ during training (x-axis) and test (y-axis), in the transductive setting (c) and the inductive setting (d). In (d) the model is tested on a new database with STUDENTS, COURSES and PROFESSORS unseen during training time. The baseline is predicting the mean value of training observations. At test time, the observed entries are used to predict the values of the fixed, held-out test set.

$\mathbf{X}^{\{d_1,d_2\}} \in \mathbb{R}^{N_{d_1} \times N_{d_2}}$ in the relational database, $\mathbb{X} = \langle \mathbf{X}^{\{1,2\}}, \mathbf{X}^{\{1,3\}}, \mathbf{X}^{\{2,3\}} \rangle$, is produced by first uniformly sampling an h-dimensional embedding for each entity instance $\mathbf{Z}^d \in \mathbb{R}^{N_d \times h}$, followed by matrix product $\mathbf{X}^{\{d_1,d_2\}} := \mathbf{Z}^{d_1}\mathbf{Z}^{d_2\top}$. A sparse subset of these matrices are observed by our model in the following experiments. Note that while we use a simple matrix product to generate the content of tables from latent factors, the model is oblivious to this generative process.

We use a *factorized auto-encoding* architecture consisting of a stack of EERN followed by pooling that produces code matrices $\mathbf{Z}^d \in \mathbb{R}^{N_d \times h'} \quad \forall d \in \{1, 2, 3\}$ for each entity, STUDENT, COURSE and PROFESSOR. The code is then fed to a decoding EERN to reconstruct the sparse $\mathrm{vec}(\mathbb{X})$. For all experiments, the encoder consists of 7 EERLs, each with 64 hidden units, each using batch normalization (Ioffe & Szegedy, 2015) and channel-wise dropout. We then apply mean pooling to produce encodings. We found that batch normalization dramatically sped up the training procedure. See Appendix H for more details.

Once we finish training the model on a relational dataset, we can apply it to another instantiation — that is a dataset with completely different STUDENTS, COURSES and PROFESSORS.

**Embedding and Missing Record Prediction.** We use a small embedding dimension $h = h' = 2$ for visualization of COURSE embeddings. Fig. 3(a) and (b) suggest that the learned embeddings agree with the ground truth. The structure within the data points has been preserved: in (a), points with a small $x$-coordinate in the ground-truth embeddings have a small $x$-coordinate in the learned embeddings (indicated by the darker color), and similarly for the $y$-coordinate and for (b). Note that in the best case, the two embeddings can agree up to a diffeomorphism. Next, we set out to predict missing records in the STUDENT-COURSE table using observed data from the whole database. For this, the factorized auto-encoding architecture is trained to only minimize the reconstruction error for "observed" entries in the STUDENT-COURSE tensor. We use $N_1 = N_2 = N_3 = 200$, with an encoding size of 10. To study the model's behaviour, we repeat the experiment while changing two quantities: 1) the percentage of observed entries (*i.e.*, sparsity of all database tensors) at training time; and 2) the sparsity of test-time observations, after the model is trained. Fig. 3(c) visualizes the average prediction error, confirming our expectation that the amount of data seen during both training and testing can increase the prediction accuracy. More importantly, once the model is trained at a particular sparsity level, it shows robustness to changes in the sparsity level at test time.

**Inductive Setting.** Once we finish training the model on a relational dataset, we can apply it to another instantiation — that is a dataset with completely different STUDENTS, COURSES and PROFESSORS. Note that this is possible because the unique values in $\mathbf{W}$ do not grow with the number of instances of each entity in our database – *i.e.*, we can resize $W$ by repeating its pattern to fit the dimensionality of our data. In practice, since we use pooling-broadcasting operations, this resizing is implicit. Fig. 3(b) reproduces the embedding experiment, and Fig. 3(d) shows the missing record

prediction results for the inductive setting. In both cases the model performs well when applied to a new database. This setting can have interesting real-world applications, as it enables transfer learning across databases and allows for predictive analysis without training for new entities in a database as they become available.

**The Value of Side Information.**  Do we gain anything by using the "entire" database for predicting missing entries of a particular table, compared to simply using the target tensor $\mathbf{X}^{\{1,2\}}$ (STUDENT-COURSE table) for both training and testing? To answer this question, we fix the sparsity level of the STUDENT-COURSE table at 0.1%, and train models with increasing levels of sparsity for other tensors $\mathbf{X}^{\{1,3\}}$ and $\mathbf{X}^{\{2,3\}}$ in the range $[0.025, 0.7]$. This figure shows that the side information in the form of STUDENT-PROFESSOR and COURSE-PROFESSOR tables can significantly improve the prediction of missing records in the STUDENT-COURSE table.

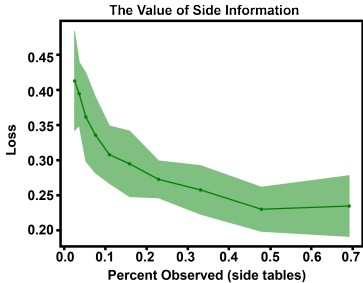

## 5.2   PREDICTING ON SOCCER DATABASE

We use the European Soccer database to build a simple relational model with three entities: PLAYER, TEAM and MATCH. The database contains information for about 11,000 PLAYERS, 300 TEAMS and 25,000 MATCHES in European soccer leagues. We extract a COMPETES-IN relation between TEAMS and MATCHES, as well as a PLAYS-IN relation between PLAYERS and MATCHES that identifies which players played in each match. Our objective is to predict whether the outcome of a match was *Home Win*, *Away Win*, or *Draw*.

A simple baseline is to always predict *Home Win*, which obtains 46% accuracy. By engineering features from temporal statistics (such as the result of recent games for a team relative to a particular target match, recent games two teams played against each other, as well as recent goal statistics) the best model reported on Kaggle achieve 56% accuracy. Without using any temporal data, by simply taking the average for any such time series, our model achieves 53% accuracy. This also matches the accuracy of professional bookies. See Appendix H for details.

## 6   RELATED LITERATURE

For a more detailed literature review see Appendix A. Our work is related to a large body of work in statistical relational learning (Getoor & Taskar, 2007; Raedt et al., 2016) and knowledge-graph (KG) embedding (Nickel et al., 2016), as well as relational (*e.g.*, Battaglia et al., 2018; Hamilton et al., 2017b), and equivariant (*e.g.*, Cohen & Welling, 2016a; Ravanbakhsh et al., 2017b; Kondor & Trivedi, 2018; Cohen et al., 2019) deep learning. When dealing with a single relation, such as STUDENT-COURSE or USER-ITEM-TAG one may use tensor factorization methods for both embedding and tensor completion. However, with more than one relation, one needs to jointly factorize a set of tensors $\mathbf{X}^{\mathbb{R}} \in \mathbb{X}$, so that the factors in common between any two tensors are in agreement – *e.g.*, in Fig. 1, $\mathbf{X}^{\{1,2\}}$ and $\mathbf{X}^{\{1,3\}}$ have a common factor, corresponding to the STUDENT entity.

Most relevant to our work in the statistical relational learning community is the relational neural network of Kazemi & Poole (2017). However, they focus on a single relation and the proposed model is more directly comparable to Hartford et al. (2018). See Appendix A.1.1 for discussion.

EERL generalizes several equivariant layers proposed for structured domains: Our model reduces to (equivariant model in) Deep-Sets (Zaheer et al., 2017) when we have a single relation $\mathbb{R} = \{1\}$ with a single entity – *i.e.*, $D = 1$; *i.e.*, a set of instances. Hartford et al. (2018) consider a more general setting of interaction across different sets, such as user-tag-movie relations. EERLs reduce to their parameter-sharing when we have a single relation $\mathfrak{R} = \{\mathbb{R}\}$ with multiple entities $D \geq 1$, where all entities appear only once. Maron et al. (2018) further relax the assumption above, and allow for multiset relations. Intuitively, this form of relational data can model the interactions within and between sets; for example interaction within nodes of a graph is captured by an adjacency matrix, corresponding to $D = 1$ and $\mathbb{R} = \{1, 1\}$. Our model reduces to this scheme when restricted to a single relation. For detailed discussion on these relations see Appendix A.

## CONCLUSION AND FUTURE WORK

We have outlined a novel and principled approach to deep learning with relational data(bases). In particular, we introduced a simple constraint in the form of tied parameters for the standard feed-forward layer and proved that any other tying scheme either ignores the exchangeabilities of relational data or can be obtained by further constraining our model. The proposed layer can be applied in inductive settings, where the relational databases used during training and test have no overlap. While our model enjoys a linear computational complexity in the size of the database, we have to overcome one more hurdle before applying this model to large-scale real-world databases: relational databases often hold large amount of data, and in order for our model to be applied in these settings we need to perform mini-batch sampling. However, any such sampling has the effect of sparsifying the observed relations. A careful sampling procedure is required that minimizes this sparsification for a particular subset of entities or relations. While several recent works propose solutions to similar problems on graphs and tensors (*e.g.*, Hamilton et al., 2017a; Hartford et al., 2018; Ying et al., 2018; Eksombatchai et al., 2017; Chen et al., 2018; Huang et al., 2018), we leave this important direction for relational databases to future work.

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

# A   A MORE DETAILED REVIEW OF RELATED LITERATURE

To our knowledge there are no similar frameworks for direct application of deep models to relational databases, and current practice is to automate feature-engineering for specific prediction tasks (Lam et al., 2018).

## A.1   STATISTICAL RELATIONAL LEARNING AND KNOWLEDGE-GRAPH EMBEDDING

Statistical relational learning extends the reach of probabilistic inference to the relational model (Raedt et al., 2016). In particular, a variety of work in *lifted inference* procedures extends inference methods in graphical models to the relational setting, where in some cases the symmetry group of the model is used to speed up inference (Kersting, 2012). Most relevant to our work from this community is the Relational Neural Network model of Kazemi & Poole (2017); see Appendix A.1.1.

An alternative to inference with symbolic representations of relational data is to use embeddings. In particular, *Tensor factorization* methods that offer tractable inference in latent variable graphical models  (Anandkumar et al., 2014), are extensively used for knowledge-graph embedding (Nickel et al., 2016). A knowledge-graph can be expressed as an ER diagram with a single relation $\mathbb{R} = \{1_{(1)}, 2, 1_{(2)}\}$, where $1_{(1)}, 1_{(2)}$ representing HEAD and TAIL entities and $2$ is an entity representing the RELATION. Alternatively, one could think of knowledge-graph as a graph representation for an instantiated ER diagram (as opposed to a set of of tables or tensors). However, in knowledge-graphs, an entity-type is a second class citizen, as it is either another attribute, or it is expressed through relations to special objects representing different "types". Therefore, compared to ER diagrams, knowledge-graphs are less structured, and more suitable for representing a variety of relations between different objects, where the distinction between entity types is not key.

### A.1.1   RELATION TO RELNN OF KAZEMI & POOLE (2017)

An alternative approach explored in the statistical relational learning community includes extensions of logistic regression to relational data (Kazemi et al., 2014), and further extensions to multiple layers (Kazemi & Poole, 2017). The focus of these works is primarily on predicting properties of the various entity instances (the example they use is predicting a user's gender based on the ratings given to movies).

Their model works by essentially counting the number of instances satisfy a given properties, but is easiest understood by interpreting it as a series of convolution operations using row- and column-wise filters that capture these properties. Consider Example 3 from (Kazemi & Poole, 2017) (also depicted in their Figure 4). We have a set of USERS and a set of MOVIES, and there is a matrix $\mathbf{L} \in \mathbb{R}^{N \times M}$, denoting which movies where liked by which users. As filters, they use binary vectors $\mathbf{a} \in \mathbb{R}^M$, and $\mathbf{o} \in \mathbb{R}^N$, representing which movies are ACTION and which users are OLD, respectively. The task they pose is to predict the gender of a user[5], given this information. To do so, they include a third filter, $\boldsymbol{\phi} \in \mathbb{R}^M$, of learnable, "numeric latent properties". Each layer of their model then convolves each of these filters with the LIKES matrix, then applies a simple linear scale and shift and a sigmoid activation. The result is three new filter vectors that can be used to make predictions or as filters in the next layer. For one layer, the outputs are then

$$\mathbf{v}^a = \sigma\left(w_0^a + w_1^a \mathbf{L}\mathbf{a}\right), \ \mathbf{v}^o = \sigma\left(w_0^o + w_1^o \mathbf{o}^T \mathbf{L}\right), \ \mathbf{v}^\phi = \sigma\left(w_0^\phi + w_1^\phi \mathbf{L}\boldsymbol{\phi}\right)$$

where each $w$ is a scalar. Observe that, for example, the $n^{\text{th}}$ element of $\mathbf{L}\mathbf{a}$ counts the number of action movies liked by user $n$. Observe also that $\mathbf{v}^a, \mathbf{v}^\phi \in \mathbb{R}^N$, while $\mathbf{v}^o \in \mathbb{R}^M$. So $\mathbf{v}^a$ and $\mathbf{v}^\phi$ can be used to make predictions about individual users. Note that the number of parameters in their model grows both with the number of movies and with the number of layers in the network.

Application of EERL to this example, reduces the 4 parameter model of (Hartford et al., 2018). Indeed, most discussions and "all" experiments in Kazemi & Poole (2017) assume a single relation. For completeness, we explain EERL in this setting. Consider the LIKES matrix, and the ACTION and

---

[5]For simplicity, and to follow the example of (Kazemi & Poole, 2017), we assume binary genders. However, we note that the real world is somewhat more complicated.

OLD filters as tables. We predict the gender of the $n^{\text{th}}$ user, as

$$\mathbf{g}_n = \sigma\left( w_0^o \mathbf{o}_n + w_1^o \sum_{n'=1}^N \mathbf{o}_{n'} + w_0^L \sum_{m=1}^M \mathbf{L}_{n,m} + w_1^L \sum_{n'=1}^M \sum_{m=1}^M \mathbf{L}_{n',m} + w_0^a \sum_{m=1}^M \mathbf{a}_m \right). \quad (5)$$

The main difference between their model and ours is that they require per-item parameters (e.g., one parameter per movie), while, as can be seen from (5), the number of parameters in our model is independent of the number of instances and so does not grow with the number of users or movies (note that we have the option of adding such features to our model by having unique one-hot features for each USER and MOVIE.) As a result, our model can be applied in inductive settings as well. One may also draw a parallel between row and column convolution in Kazemi & Poole (2017) with two out of four pooling operations when we have single relation between two entities. However these operations become insufficient when moving to models of self-relation (*e.g.*, 15 parameters for a single self-relation) and does cannot adequately capture the interaction between multiple relations as discussed in our provably optimal linear layer.

## A.2 RELATIONAL, GEOMETRIC AND EQUIVARIANT DEEP LEARNING

See Hamilton et al. (2017b); Battaglia et al. (2018) for a detailed review. Scarselli et al. (2009) introduced a generic framework that iteratively updates node embeddings using neural networks. Li et al. (2015) integrated this iterative process in a recurrent architecture. Gilmer et al. (2017) proposed a similar iterative procedure that updates node embeddings and messages between the neighbouring nodes, and show that it subsumes several other deep models for attributed graphs (Duvenaud et al., 2015; Schütt et al., 2017; Li et al., 2015; Battaglia et al., 2016; Kearnes et al., 2016), including spectral methods. Their method is further generalized in (Kondor et al., 2018b) as well as (Maron et al., 2018), which is in turn subsumed in our framework. Spectral methods extend convolution to graphs (and manifolds) using eigenvectors of the Laplacian as the generalization of the Fourier basis (Bronstein et al., 2017; Bruna et al., 2014). Simplified variations of this approach leads to an intuitive yet non-maximal parameter-sharing scheme that is widely used in practice (Kipf & Welling, 2016). This type of simplified graph convolution has also been used for relational reasoning with knowledge-graphs (Schlichtkrull et al., 2018).

An alternative generalization of convolution is defined for functions over groups (Cohen & Welling, 2016a) or more generally homogeneous spaces (Cohen et al., 2018b). Moreover, convolution can be performed in the Fourier domain in this setting, where irreducible representations of a group become the Fourier bases (Kondor & Trivedi, 2018). Particularly relevant to our work are the models of (Kondor et al., 2018b) and (Maron et al., 2018) that operate on graph data using an equivariant design. Equivariant deep models for a variety of structured domains is explored in several other recent works. (*e.g.*, Worrall et al., 2017; Cohen et al., 2018a; Kondor et al., 2018a; Sabour et al., 2017; Weiler et al., 2018); see also (Cohen & Welling, 2016b; Weiler et al., 2017; Kondor et al., 2018b; Anselmi et al., 2019).

## A.3 PARAMETER-SHARING, EXCHANGEABILITY AND EQUIVARIANCE

The notion of invariance is also studied under the term exchangeability in statistics (Orbanz & Roy, 2015); see also (Bloem-Reddy & Whye Teh, 2019) for a probabilistic approach to equivariance. In graphical models exchangeability is often encoded through plate notation, where parameter-sharing happens implicitly. In the AI community, this relationship between the parameter sharing and "invariance" properties of the network was noticed in the early days of the Perceptron (Minsky & Papert, 2017; Shawe-Taylor, 1989; 1993). This was rediscovered in (Ravanbakhsh et al., 2017b), where this relation was leveraged for equivariant model design. Ravanbakhsh et al. (2017b) show that when the group action is discrete "equivariance" to any group action can be obtained by parameter-sharing. In particular, one of the procedures discussed there ties the elements of $\mathbf{W}$ based on the *orbits* of the "joint" action of the group on the rows and columns of $\mathbf{W}$. The parameter-tying scheme in the following can be obtained in this way:

### A.3.1 RELATION TO DEEP-SETS OF ZAHEER ET AL. (2017)

propose an equivariant model for set data. Our model reduces to their parameter-tying when we have a single relation $\mathbb{R} = \{1\}$ with a single entity – *i.e.*, $D = 1$; *i.e.*, a set of instances; see also Example 3 in Appendix E. Since we have a single relation, $\mathbf{W}$ matrix has a single block $\mathbf{W} = \mathbf{W}^{1,1}$, indexed by $n^1$. The elements of $\mathbf{n}^{1,1} = \mathbf{n}_1^{1,1} = \langle n_{1_{(1)}}, n_{1_{(2)}} \rangle$ index the elements of this matrix, for entity 1 (the only entity). There are two types of equality patterns $n_{1_{(1)}} = n_{1_{(2)}}$, and $n_{1_{(1)}} \neq n_{1_{(2)}}$, giving rise to the permutation equivariant layer introduced in (Ravanbakhsh et al., 2017a; Zaheer et al., 2017).

### A.3.2 RELATIONP TO EXCHANGEABLE TENSOR MODELS OF HARTFORD ET AL. (2018)

Hartford et al. (2018) consider a more general setting of interaction across different sets, such as user-tag-movie relations. Our model produces their parameter-sharing when we have a single relation $\mathfrak{R} = \{\mathbb{R}\}$ with multiple entities $D \geq 1$, where all entities appear only once – *i.e.*, $\kappa(d) = 1 \forall d \in \mathbb{R}$. Here, again $\mathbf{W} = \mathbf{W}^{1,1}$, and $\mathbf{n}^{1,1}$, the concatenation of row-index $\mathbf{n}^1$ and column index $\mathbf{n}^1$, identifies an element of this matrix. Since each $d \in \mathbb{R}$ has multiplicity 1, $\mathbf{n}_d^{1,1} = \langle \mathbf{n}_{d_{(1)}}^{1,1}, \mathbf{n}_{d_{(2)}}^{1,1} \rangle \ \forall d \in [D]$, and therefore $\mathbf{n}_d^{1,1}$ can have two class of equality patterns. This gives $2^D$ equivalence classes for $\mathbf{n}^{1,1}$, and therefore $2^D$ unique parameters for a rank $D$ exchangeable tensor.

### A.3.3 RELATIONSHIP TO EQUIVARIANT GRAPH NETWORKS OF MARON ET AL. (2018)

Maron et al. (2018) further relax the assumption of Hartford et al. (2018), and allow for $\kappa(d) \geq 1$. Intuitively, this form of relational data can model the interactions within and between sets; for example interaction within nodes of a graph is captured by an adjacency matrix, corresponding to $D = 1$ and $\mathbb{R} = \{1, 1\}$. This type of parameter-tying is maximal for graphs, and subsumes the parameter-tying approaches derived by simplification of Laplacian-based methods. When restricted to a single relation, our model reduces to the model of (Maron et al., 2018); however, when we have multiple relations, $\mathbf{W}^{i,j}$ for $j \neq i$, our model captures the interaction between different relations / tensors.

| | | | | |
|---|---|---|---|---|
| $\mathbf{x}, \mathbf{n}, \mathbf{m}$ | tuple or column vector (bold lower-case) | | $\text{vec}(\mathbb{X})$ | $\langle \text{vec}(\mathbf{X}^1), \ldots, \text{vec}(\mathbf{X}^{|\mathfrak{R}|}) \rangle$ |
| $\langle \cdot, \cdot \rangle$ | a tuple | | $N^i = \prod_{d \in \mathbb{R}_i} N_d$ | length of $\text{vec}(\mathbf{X}^i)$ |
| $\mathbf{X}, \mathbf{G}, \mathbf{W}$ | tensor, inc. matrix (bold upper-case) | | $N = \sum_{\mathbb{R}_i \in \mathfrak{R}} N^i$ | length of $\text{vec}(\mathbb{X})$ |
| $\mathbb{X}, \mathbb{R}$ | set (or multiset) | | $\mathbf{W} \in \mathbb{R}^{N \times N}$ | parameter matrix |
| $\mathcal{S}, \mathcal{G}$ | group (caligraphic) | | $\mathbf{W}^{i,j} \in \mathbb{R}^{N^i \times N^j}$ | $(i,j)^{th}$ block of $\mathbf{W}$ |
| $\mathscr{D} = [D] = \{1, \ldots, D\}$ | set of entities | | $\mathbf{n}^i = \langle n_{d_1}^i, \ldots, n_{d_{|\mathbb{R}_i|}}^i \rangle$ | index for $\text{vec}(\mathbf{X}^i)$ |
| $N_1, \ldots, N_D$ | number of instances | | | and for rows of $\mathbf{W}^{i,j}$ |
| $\mathfrak{R} \subseteq 2^{\mathscr{D}}$ | a set of relations | | $\mathbf{n} = \langle \mathbf{n}^1, \ldots, \mathbf{n}^{|\mathfrak{R}|} \rangle$ | index for $\text{vec}(\mathbb{X})$ |
| $\mathbb{R}_i \subseteq \mathscr{D}$ | a relation | | | and rows/columns of $\mathbf{W}$ |
| $\mathbf{X}^i = \mathbf{X}^{\mathbb{R}_i} \in \mathbb{R}^{N_{d_1} \times \ldots, N_{d_{|\mathbb{R}_i|}}}$ | data for a relation $\mathbb{R}_i$ | | $\mathcal{S}^M$ | symmetric group $M$ |
| $\mathbb{X} = \{\mathbf{X}^i \mid \mathbb{R}_i \in \mathfrak{R}\}$ | relational data | | $\mathcal{G}^{\mathbb{X}}$ | group of $N \times N$ "legal" |
| $\text{vec}(\mathbf{X}^i)$ | vectorization of $\mathbf{X}^i$ | | | permutations of $\text{vec}(\mathbb{X})$ |

Table 1: Summary of Notation

## B  MULTISET RELATIONS

Because we allow a relation $\mathbb{R}$ to contain the same entities multiple times, we formally define a multiset as a tuple $\widetilde{\mathbb{R}} = \langle \mathbb{R}, \kappa \rangle$, where $\mathbb{R}$ is a set, and $\kappa : \mathbb{R} \to \mathbb{N}$ maps elements of $\mathbb{R}$ to their multiset counts. We will call $\mathbb{R}$ the *elements* of the multiset $\widetilde{\mathbb{R}} = \langle \mathbb{R}, \kappa \rangle$, and $\kappa(d)$ the *count* of element $d$. We define the union and intersection of two multisets $\mathbb{R}_1$ and $\mathbb{R}_2$ as $\widetilde{\mathbb{R}}_1 \cup \widetilde{\mathbb{R}}_2 = \langle \mathbb{R}_1 \cup \mathbb{R}_2, \kappa_1 + \kappa_2 \rangle$ and $\widetilde{\mathbb{R}}_1 \cap \widetilde{\mathbb{R}}_2 = \langle \mathbb{R}_1 \cap \mathbb{R}_2, \min\{\kappa_1, \kappa_2\} \rangle$. In general, we may also refer to a multiset using typical set notation (*e.g.*, $\mathbb{R} = \{d_1, d_1, d_2\}$). We will use bracketed superscripts to distinguish distinct but equal members of any multiset (*e.g.*, $\mathbb{R} = \{d_1, d_1, d_2\} = \{d_1^{(1)}, d_1^{(2)}, d_2^{(1)}\}$). The ordering of equal members is specified by context or arbitrarily. The size of a multiset $\widetilde{\mathbb{R}}$ accounts for multiplicities: $|\widetilde{\mathbb{R}}| = \sum_{d \in \mathbb{R}} \kappa(d)$.

## C  NUMBER OF FREE PARAMETERS

For the multiset relations $\widetilde{\mathbb{R}_i} = \langle \mathbb{R}_i, \kappa \rangle$ and $\widetilde{\mathbb{R}_j} = \langle \mathbb{R}_j, \kappa \rangle$, recall that two parameters $\mathbf{W}^{i,j}_{\mathbf{n}^i, \mathbf{n}^j}$ and $\mathbf{W}^{i,j}_{\mathbf{m}^i, \mathbf{m}^j}$ are tied together if $\mathbf{n}^{i,j}$, the concatenation of $\mathbf{n}^i$ with $\mathbf{n}^j$, is in the same equivalence class as $\mathbf{m}^{i,j}$. We partition each $\mathbf{n}^{i,j}_d$ into sub-partitions $\mathbb{P}(\mathbf{n}^{i,j}_d) \doteq \{\mathbb{P}_1, \ldots, \mathbb{P}_L\}$ of indices whose values are equal, and consider $\mathbf{n}^{i,j}$ and $\mathbf{m}^{i,j}$ to be equivalent if their partitions are the same for all $d$:

$$\mathbf{n}^{i,j} \equiv \mathbf{m}^{i,j} \Leftrightarrow \mathbb{P}(\mathbf{n}^{i,j}_d) = \mathbb{P}(\mathbf{m}^{i,j}_d) \; \forall d \in \mathbb{R}_i \cup \mathbb{R}_j \tag{6}$$

See 4.1 for details. This means that the total number of *free* parameters in $\mathbf{W}^{i,j}$ is the product of the number of possible different partitionings for each unique entity $d \in \mathbb{R}_i \cup \mathbb{R}_j$:

$$|\mathbf{w}^{i,j}| = \prod_{d \in \mathbb{R}_i \cup \mathbb{R}_j} b(\kappa^i(d) + \kappa^j(d)) \tag{7}$$

where $\mathbf{w}^{i,j}$ is the free parameter vector associated with $\mathbf{W}^{i,j}$, and $b(\kappa)$ is the $\kappa^{th}$ Bell number, which counts the possible partitionings of a set of size $\kappa$.

**Example 2.** *[Fig. 2(a)] Consider again the simplified version of the relational structure of Fig. 1, restricted to the three relations $\mathbb{R}_1 = \{1, 2\}$, self-relation $\mathbb{R}_2 = \{2, 2\}$, and $R_3 = \{1, 3\}$ with $N_1 = 5$ STUDENTS, $N_2 = 4$ COURSES, and $N_3 = 3$ PROFESSORS. We use tuple $\mathbf{n}^1 = \langle n^1_1, n^1_2 \rangle \in [N_1] \times [N_2]$ to index the rows and columns of $\mathbf{W}^{1,1}$. We also use $\mathbf{n}^1$ to index the rows of $\mathbf{W}^{1,2}$, and use $\mathbf{n}^2 = \langle n^2_{2(1)}, n^2_{2(2)} \rangle \in [N_2] \times [N_2]$ to index its columns. Other blocks are indexed similarly. The elements of $\mathbf{W}^{1,1}$ take $b(2)b(2) = 4$ different values, depending on whether or not $n^1_1 = {n^{1'}_1}$ and $n^1_2 = {n^{1'}_2}$, for row index $\mathbf{n}^1$ and column index $\mathbf{n}^{1'}$ (where $b(k)$ is the k-th Bell number). The elements of $\mathbf{W}^{1,2}$ take $b(1)b(3) = 5$ different values: the index $n^1_1$ can only be partitioned in a single way ($b(1) = 1$). However index $n^1_2$ and indices $n^2_{2(1)}$ and $n^2_{2(2)}$ all index into the COURSES table, and so can each potentially refer to the same course. We thus have a unique parameter for each possible combination of equalities between these three items, giving us a factor of $b(3) = 5$ different parameter values; see Fig. 2(a), $\mathbf{W}^{1,1}$ is the upper left block, and $\mathbf{W}^{1,2}$ is the block to its right. The center block of Fig. 2(a), $\mathbf{W}^{2,2}$ produces the effect of $\mathbb{R}_2 = \{2, 2\}$ on itself. Here, all four index values could refer to the same course, and so there are $b(4) = 15$ different parameters.*

## D  BIAS PARAMETERS

For full generality, our definition of EERL could also include bias terms without affecting its exchangeability properties. We exclude these in the statements of our main theorems for the sake of simplicity, but discuss their inclusion here for completeness. For each relation $\mathbb{R}_i = \{d_1, ..., d_{|\mathbb{R}_i|}\}$, we define a bias tensor $\mathbf{B}^i \in \mathbb{R}^{N_{d_1} \times ... \times N_{d_{|\mathbb{R}_i|}}}$. The elements of $\mathbf{B}^i$ are tied together in a manner similar to the tying of elements in each $\mathbf{W}^{i,j}$: Two elements $\mathbf{B}^i_{\mathbf{n}^i}$ and $\mathbf{B}^i_{\mathbf{m}^i}$ are tied together iff $\mathbf{n}^i \equiv \mathbf{n}^j$, using the definition of equivalence from Section 4.1. Thus, we have a vector of additional free parameters $\mathbf{b}^i$ for each relation $\mathbb{R}_i$, where

$$|\mathbf{b}^i| = \prod_{d \in \mathbb{R}_i} b(\kappa^i(d)). \tag{8}$$

Consistent with our previous notation, we define $\mathbb{B} = \{\mathbf{B}^i \mid \mathbb{R}_i \in \mathfrak{R}\}$, and $\mathrm{vec}(\mathbb{B}) = \langle \mathrm{vec}(\mathbf{B}^1), ..., \mathrm{vec}(\mathbf{B}^{|\mathfrak{R}|}) \rangle$. Then an EERL with bias terms is given by

$$\mathrm{vec}(\mathbf{Y}) = \sigma\big(\mathbf{W} \, \mathrm{vec}(\mathbb{X}) + \mathrm{vec}(\mathbb{B})\big). \tag{9}$$

The following Claim asserts that we can add this bias term without affecting the desired properties of the EERL.

**Claim 2.** *If $\sigma\big(\mathbf{W} \, \mathrm{vec}(\mathbb{X})\big)$ is an EERL, then $\sigma\big(\mathbf{W} \, \mathrm{vec}(\mathbb{X}) + \mathrm{vec}(\mathbb{B})\big)$ is an EERL.*

The proof (found in Appendix G.1) argues that, since $\sigma\big(\mathbf{W} \, \mathrm{vec}(\mathbb{X})\big)$ is an EERL, we just need to show that $\mathbf{G}^{\mathbb{X}} \, \mathrm{vec}(\mathbb{B}) = \mathrm{vec}(\mathbb{B})$ iff $\mathbf{G}^{\mathbb{X}} \in \mathcal{G}^{\mathbb{X}}$, which holds due to the tying of patterns in each $\mathbf{B}^i$.

# E   SIMPLIFICATIONS FOR MODELS WITHOUT SELF-RELATIONS

In the special case that the multi relations $\widetilde{\mathbb{R}}_i$ and $\widetilde{\mathbb{R}}_j$ are sets —*i.e.*, have no self-relations— then the parameter tying scheme of Section 4.1 can be simplified considerably. In this section we address some nice properties of this special setting.

## E.1   EFFICIENT IMPLEMENTATION USING SUBSET-POOLING

Due to the particular structure of $\mathbf{W}$ when all relations contain only unique entities, the operation $\mathbf{W}\,\text{vec}(\mathbb{X})$ in the EERL can be implemented using (sum/mean) pooling operations over the tensors $\mathbf{X}^{\mathbb{R}}$ for $\mathbb{R} \in \mathfrak{R}$, without any need for vectorization, or for storing $\mathbf{W}$ directly.

For $\mathbf{X}^i \in \mathbb{R}^{N_{d_1} \times \ldots \times N_{d_{|\mathbb{R}|}}}$ and $\mathbb{S} \subseteq \mathbb{R}_i$, let $\text{pool}(\mathbf{X}^i, \mathbb{S})$ be the summation of the tensor $\mathbf{X}^i$ over the dimensions specified by $\mathbb{S}$. That is, $\text{pool}(\mathbf{X}^i, \mathbb{S}) \in \mathbb{R}^{N_{d'_1} \times \ldots \times N_{d'_{|\mathbb{R}_i| - |\mathbb{S}|}}}$ where $\mathbb{R}_i \backslash \mathbb{S} = \{d'_1, \ldots, d'_{|\mathbb{R}|-|\mathbb{S}|}\}$. Then we can write element $\mathbf{n}^i$ in the $i$-th block of $\mathbf{W}\,\text{vec}(\mathbb{X})$ as

$$(\mathbf{W}\,\text{vec}(\mathbb{X}))_{(i,\mathbf{n}^i)} = \sum_{\mathbb{R}_j \in \mathfrak{R}} \sum_{\mathbb{S} \subseteq \mathbb{R}_i \cap \mathbb{R}_j} w_{\mathbb{S}}^{i,j} \, \text{pool}(\mathbf{X}^j, \mathbb{S})_{\mathbf{n}_{\mathbb{R}_i \backslash \mathbb{S}}^i} \tag{10}$$

where $\mathbf{n}_{\mathbb{R}_i \backslash \mathbb{S}}^i$ is the restriction of $\mathbf{n}^i$ to only elements indexing entities in $\mathbb{R}_i \backslash \mathbb{S}$. This formulation lends itself to a practical, efficient implementation where we simply compute each $\text{pool}(\mathbf{X}^i, \mathbb{S})$ term and broadcast-add them back into a tensor of appropriate dimensions.

## E.2   ONE-TO-ONE AND ONE-TO-MANY RELATIONS

In the special case of a one-to-one or one-to-many relations (*e.g.*, in Fig. 1, one PROFESSOR may teach many COURSES, but each COURSE has only one PROFESSOR), we may further reduce the number of parameters due to redundancies. Suppose $\mathbb{R}_i \in \mathfrak{R}$ is some relation, and entity $d \in \mathbb{R}_i$ is in a one-to- relation with the remaining entities of $\mathbb{R}_i$. Consider the 1D sub-array of $\mathbb{X}^{\mathbb{R}_i}$ obtained by varying the value of $\mathbf{n}_d^i$ while holding the remaining values $\mathbf{n}_{\mathbb{R}_i \backslash \{d\}}^i$ fixed. This sub-array contains just a single non-zero entry. According to the tying scheme described in Section 4.1, the parameter block $\mathbf{W}^{i,i}$ will contain unique parameter values $w_{\mathbb{R}_i}$ and $w_{\mathbb{R}_i \backslash \{d\}}$. Intuitively however, these two parameters capture exactly the same information, since the sub-array obtained by fixing the values of $\mathbb{R}_i \backslash \{d\}$ contains exactly the same data as the sub-array obtained by fixing the values of $\mathbb{R}_i$ (*i.e.*, the same single value). More concretely, to use the notation of Appendix E.1, we have $\text{pool}(\mathbf{X}^i, \mathbb{R}_i \backslash \{d\})_{\mathbf{n}_{\{d\}}^i} = \text{pool}(\mathbf{X}^i, \mathbb{R}_i)$ in (10), and so we may tie $w_{\mathbb{R}_i}^{i,i}$ and $w_{\mathbb{R}_i \backslash \{d\}}^{i,i}$.

In fact, we can reduce the number of free parameters in the case of self-relations (*i.e.*, relations with non-unique entities) as well in a similar manner.

## E.3   RECURSIVE DEFINITION OF THE WEIGHT MATRIX

We are able to describe the form of the parameter matrix $\mathbf{W}^{i,j}$ concisely in a recursive fashion, using Kronecker products: For any $N \in \mathbb{N}$, let $\mathbf{1}_N \in \mathbb{R}^{N \times N}$ be the $N \times N$ matrix of all ones, and $\mathbf{I}_N$ the $N \times N$ identity matrix. Given any set of (unique) entities $\mathbb{S} = \{d_1, \ldots, d_{|\mathbb{S}|}\} \subseteq \{1, \ldots, D\}$, for $k = 1, \ldots, |\mathbb{S}|$, recursively define the sets

$$\mathbb{W}_k^{\mathbb{S}} = \left\{ \mathbf{W} \otimes \mathbf{1}_{N_{d_k}} + \mathbf{V} \otimes \mathbf{I}_{N_{d_k}} \mid \mathbf{W}, \mathbf{V} \in \mathbb{W}_{k-1}^{\mathbb{S}} \right\}, \tag{11}$$

with the base case of $\mathbb{W}_0^{\mathbb{S}} = \mathbb{R}$. Then for each block $\mathbf{W}^{i,j}$ of (4) we simply have $\mathbf{W}^{i,j} \in \mathbb{W}_{|\mathbb{R}_i \cap \mathbb{R}_j|}^{\mathbb{R}_i \cap \mathbb{R}_j}$.

Writing the blocks of the matrix (4) in this way makes it clear why block $\mathbf{W}^{i,j}$ contains $2^{|\mathbb{R}_i \cap \mathbb{R}_j|}$ unique parameter values in the case of distinct entities: at each level of the recursive definition we are doubling the total number of parameters by including terms from two elements from the level below. It also makes it clear that the parameter matrix for a rank-$k$ tensor is built from copies of parameter matrices for rank-$(k-1)$ tensors.

**Example 3.** *In the simple case where we have just one relation and one entity, $\mathbb{R} = \{d\}$ and the parameter matrix is an element of $\mathbb{W}_1^{\mathbb{R}} = \{w \otimes \mathbf{1}_{N_d} + v \otimes \mathbf{I}_{N_d} \mid w, v \in \mathbb{R}\}$, which matches the parameter tying scheme of Zaheer et al. (2017). If instead we have a single relation with two distinct entities $\mathbb{R} = \{d_1, d_2\}$, then the parameter matrix is an element of $\mathbb{W}_2^{\mathbb{R}} = \{\mathbf{W} \otimes \mathbf{1}_{N_{d_2}} + \mathbf{V} \otimes \mathbf{I}_{N_{d_2}} \mid \mathbf{W}, \mathbf{V} \in \mathbb{W}_1^{R}\}$, which matches the tying scheme of Hartford et al. (2018).*

## F  USING MULTIPLE CHANNELS

Equivariance is maintained by composition of equivariant functions. This allows us to stack EERLs to build "deep" models that operate on relational databases. Using multiple input ($K$) and output ($K'$) channels is also possible by replacing the parameter matrix $\mathbf{W} \in \mathbb{R}^{N \times N}$, with the parameter tensor $\mathbf{W} \in \mathbb{R}^{K \times N \times N \times K'}$; while $K \times K'$ copies have the same parameter-tying pattern —*i.e.*, there is no parameter-sharing "across" channels. The single-channel matrix-vector product in $\sigma(\mathbf{W} \operatorname{vec}(\mathbb{X}))$ where $(\mathbf{W} \operatorname{vec}(\mathbb{X}))_{\mathbf{n}} = \sum_{\mathbf{n}'} \mathbf{W}_{\mathbf{n},\mathbf{n}'} \operatorname{vec}(\mathbb{X})_{\mathbf{n}'}$ is now replaced with contraction of two tensors $(\mathbf{W} \operatorname{vec}(\mathbb{X}))_{\mathbf{n},k'} = \sum_{\mathbf{n}',k \in [K]} \mathbf{W}_{k,\mathbf{n},\mathbf{n}',k'} \operatorname{vec}(\mathbb{X})_{\mathbf{n}',k}$, for $k' \in [K']$.

## G  PROOFS

Observe that for any index tuple $\mathbf{n}$, we can express $\mathbb{P}(\mathbf{n}_d)$ (Section 4.1) as

$$\mathbb{P}(\mathbf{n}_d) = \left\{ \left\{ d_{(k)}, d_{(l)} \; \forall k, l \mid n_{d_{(k)}} = n_{d_{(l)}} \right\} \;\middle|\; d_{(k)} = d_{(l)} = d \right\}. \tag{12}$$

We will make use of this formulation in the proofs below.

### G.1  PROOF OF CLAIM 2

*Proof.* We want to show that

$$\mathbf{G}^{\mathbb{X}} \sigma\big(\mathbf{W} \operatorname{vec}(\mathbb{X}) + \operatorname{vec}(\mathbb{B})\big) = \sigma\big(\mathbf{W} \mathbf{G}^{\mathbb{X}} \operatorname{vec}(\mathbb{X}) + \operatorname{vec}(\mathbb{B})\big) \tag{13}$$

iff $\mathbf{G}^{\mathbb{X}} \in \mathcal{G}^{\mathbb{X}}$. Since $\sigma\big(\mathbf{W} \operatorname{vec}(\mathbb{X})\big)$ is an EERL, this is equivalent to showing

$$\mathbf{G}^{\mathbb{X}} \operatorname{vec}(\mathbb{B}) = \operatorname{vec}(\mathbb{B}) \iff \mathbf{G}^{\mathbb{X}} \in \mathcal{G}^{\mathbb{X}}. \tag{14}$$

($\impliedby$) Suppose $\mathbf{G}^{\mathbb{X}} \in \mathcal{G}^{\mathbb{X}}$, with $\mathcal{G}^{\mathbb{X}}$ defined as in Claim 1. Fix some relation $\widetilde{\mathbb{R}}_i$ and consider the $i$-th block of $\mathbf{G}^{\mathbb{X}} \operatorname{vec}(\mathbb{B})$:

$$\big(\mathbf{G}^{\mathbb{X}} \operatorname{vec}(\mathbb{B})\big)_{(i,\mathbf{n}^i)} = \big(\mathbf{K}^i \operatorname{vec}(\mathbf{B}^i)\big)_{\mathbf{n}^i} \tag{15}$$

$$= \sum_{\mathbf{n}^{i'}} \mathbf{K}^i_{\mathbf{n}^i, \mathbf{n}^{i'}} \operatorname{vec}(\mathbf{B}^i)_{\mathbf{n}^{i'}} \tag{16}$$

$$= \operatorname{vec}(\mathbf{B}^i)_{\mathbf{n}^{i*}}, \tag{17}$$

Where $\mathbf{G}^{\mathbb{X}} = \operatorname{diag}(\mathbf{K}^1, ..., \mathbf{K}^{|\mathfrak{R}|})$, and $\mathbf{n}^{i*}$ is the unique index such that $\mathbf{K}^i_{\mathbf{n}^i, \mathbf{n}^{i*}} = 1$. As above, let $\mathbf{n}^i_d$ be the restriction of $\mathbf{n}^i$ to elements indexing entity $d$. Then we want to show that $\mathbb{P}(\mathbf{n}^i_d) = \mathbb{P}(\mathbf{n}^{i*}_d)$ for all $d \in \mathbb{R}_i$. Now, since $\mathbf{G}^{\mathbb{X}} \in \mathcal{G}^{\mathbb{X}}$ and $\mathbf{K}^i_{\mathbf{n}^i, \mathbf{n}^{i*}} = 1$ we have

$$\mathbf{G}^d_{n^i_{d_{(k)}}, n^{i*}_{d_{(k)}}} = 1 \tag{18}$$

for all $d \in \mathbb{R}_i$ and all $k$. That is $g^d(n^{i*}_{d_{(k)}}) = n^i_{d_{(k)}}$. Consider $\mathbb{S} \in \mathbb{P}(\mathbf{n}^i_d)$. we have

$$\mathbb{S} = \{d_{(k)}, d_{(l)} \forall k, l \mid n^i_{d_{(k)}} = n^i_{d_{(l)}}\} \tag{19}$$

$$= \{d_{(k)}, d_{(l)} \forall k, l \mid g^{d^{-1}}(n^i_{d_{(k)}}) = g^{d^{-1}}(n^i_{d_{(l)}})\} \tag{20}$$

$$= \{d_{(k)}, d_{(l)} \forall k, l \mid n^{i*}_{d_{(k)}} = n^{i*}_{d_{(l)}}\}. \tag{21}$$

So $\mathbb{P}(\mathbf{n}_d^i) \subseteq \mathbb{P}(\mathbf{n}_d^{i^*})$. A similar argument has $\mathbb{P}(\mathbf{n}_d^i) \supseteq \mathbb{P}(\mathbf{n}_d^{i^*})$. Thus, we have $\text{vec}(\mathbf{B}^i)_{\mathbf{n}^{i^*}} = \text{vec}(\mathbf{B}^i)_{\mathbf{n}^i}$, which completes the first direction.

($\Longrightarrow$) Let $\mathbf{G}^{\mathbb{X}} \text{vec}(\mathbb{B}) = \text{vec}(\mathbb{B})$. First, suppose for the sake of contradiction that $\mathbf{G}^{\mathbb{X}} \notin \bigoplus_{\mathbb{R} \in \mathfrak{R}} \mathcal{S}^{N_{\mathbb{R}}}$ and consider dividing the rows and columns of $\mathbf{G}^{\mathbb{X}}$ into blocks that correspond to each relation $\mathbb{R}_i$. Then since $\mathbf{G}^{\mathbb{X}} \notin \bigoplus_{\mathbb{R} \in \mathfrak{R}} \mathcal{S}^{N_{\mathbb{R}}}$, there exist $\mathbb{R}_i, \mathbb{R}_j \in \mathfrak{R}$, with $i \neq j$ and $\mathbf{n}^i \in [N_{d_1^i}] \times ... \times [N_{d_{|\mathbb{R}_i|}^i}]$ and $\mathbf{n}^j \in [N_{d_1^j}] \times ... \times [N_{d_{|\mathbb{R}_j|}^j}]$ such that $\mathbf{G}^{\mathbb{X}}$ maps $(i, \mathbf{n}^i)$ to $(j, \mathbf{n}^j)$. That is $g^{\mathbb{X}}\big((i, \mathbf{n}^i)\big) = (j, \mathbf{n}^j)$ and thus $\mathbf{G}^{\mathbb{X}}_{(j, \mathbf{n}^j),(i, \mathbf{n}^i)} = 1$. So

$$\big(\mathbf{G}^{\mathbb{X}} \text{vec}(\mathbb{B})\big)_{(j, \mathbf{n}^j)} = \sum_k \sum_{\mathbf{n}^k} \mathbf{G}^{\mathbb{X}}_{(j, \mathbf{n}^j),(k, \mathbf{n}^k)} \text{vec}(\mathbb{B})_{(k, \mathbf{n}^k)} \tag{22}$$

$$= \text{vec}(\mathbb{B})_{(i, \mathbf{n}^i)} \tag{23}$$

$$= \text{vec}(\mathbf{B}^i)_{\mathbf{n}^i} \tag{24}$$

$$\neq \text{vec}(\mathbf{B}^j)_{\mathbf{n}^j}, \tag{25}$$

by the definition of $\mathbb{B}$. And so $\mathbf{G}^{\mathbb{X}} \text{vec}(\mathbb{B}) \neq \text{vec}(\mathbb{B})$.

Next, suppose $\mathbf{G}^{\mathbb{X}} \in \bigoplus_{\mathbb{R} \in \mathfrak{R}} \mathcal{S}^{N_{\mathbb{R}}}$. Then for all $\mathbb{R}_i$, there exist $\mathbf{K}^i \in \mathcal{S}^{N^i}$ such that $\mathbf{G}^{\mathbb{X}} = \text{diag}(\mathbf{K}^1, ..., \mathbf{K}^{|\mathbb{R}_i|})$. For any $\mathbf{n}^i$ we have

$$\text{vec}(\mathbf{B}^i)_{\mathbf{n}^i} = \big(\mathbf{K}^i \text{vec}(\mathbf{B}^i)\big)_{\mathbf{n}^i} \tag{26}$$

$$= \sum_{\mathbf{n}^{i\prime}} \mathbf{K}^i_{\mathbf{n}^i, \mathbf{n}^{i\prime}} \text{vec}(\mathbf{B}^i)_{\mathbf{n}^{i\prime}} \tag{27}$$

$$= \text{vec}(\mathbf{B}^i)_{\mathbf{n}^{i^*}} \tag{28}$$

Where $\mathbf{n}^{i^*}$ is the unique index such that $\mathbf{K}^i_{\mathbf{n}^i, \mathbf{n}^{i^*}} = 1$. That is $k^i$ maps $\mathbf{n}^{i^*}$ to $\mathbf{n}^i$. Then by the definition of $\mathbf{B}^i$ we have

$$\mathbb{P}(\mathbf{n}_d^{i^*}) = \mathbb{P}(\mathbf{n}_d^i)$$

$$= \mathbb{P}\big((k^i(\mathbf{n}^{i^*}))_d\big) \tag{29}$$

for all $d \in \mathbb{R}^i$. (29) says that for each $d$ the action of $\mathbf{K}^i$ on elements of $\mathbf{n}^{i^*}$ is determined only by the values of those elements, not by the values of elements indexing other entities, and so $\mathbf{K}^i \in \bigotimes_{d \in \mathbb{R}_i} \mathcal{S}^{N_d}$. But (29) also says that for all $k, l$

$$n_{d_{(k)}}^{i^*} = n_{d_{(l)}}^{i^*} \iff \big(k^i(\mathbf{n}^{i^*})\big)_{d_{(k)}} = \big(k^i(\mathbf{n}^{i^*})\big)_{d_{(l)}}, \tag{30}$$

which says that the action of $\mathbf{K}^i$ is the same across any duplications of $d$ (i.e., $d_{(k)}$ and $d_{(l)}$), and so $\mathbf{K}^i = \bigotimes_{d \in \mathbb{R}_i} \mathbf{G}^d$, for some fixed $\mathbf{G}^d$, and therefore $\mathbf{G}^{\mathbb{X}} \in \mathcal{G}^{\mathbb{X}}$.

$\square$

## G.2 LEMMA 1 AND PROOF

To prove our main results about the optimality of EERLs we require the following Lemma.

**Lemma 1.** *For any permutation matrices $\mathbf{K}^i \in \mathcal{S}^{N^i}$ and $\mathbf{K}^j \in \mathcal{S}^{N^j}$ we have*

$$\mathbf{K}^i \mathbf{W}^{i,j} = \mathbf{W}^{i,j} \mathbf{K}^j \Leftrightarrow \mathbf{K}^i = \bigotimes_{d \in \widetilde{\mathbb{R}}_i} \mathbf{G}^d \text{ and } \mathbf{K}^j = \bigotimes_{d \in \widetilde{\mathbb{R}}_j} \mathbf{G}^d \quad \mathbf{G}^d \in \mathcal{S}^{N_d}$$

*for constrained $\mathbf{W}^{i,j}$ as define above. That is $\mathbf{K}^i$ and $\mathbf{K}^j$ should separately permute the instances of each entity in the multisets $\widetilde{\mathbb{R}}_i$ and $\widetilde{\mathbb{R}}_j$, applying the same permutation to any duplicated entities, as well as to any entities common to both $\widetilde{\mathbb{R}}_i$ and $\widetilde{\mathbb{R}}_j$.*

To get an intuition for this lemma, consider the special case of $i = j$. In this case, the claim is that $\mathbf{W}^{i,i}$ commutes with any permutation matrix that is of the form $\mathbf{K}^i = \bigotimes_{d \in \mathbb{R}_i} \mathbf{G}^d$. This gives us the kind of commutativity we desire for an EERL, at least for the diagonal blocks of $\mathbf{W}$. Equivalently, commuting with $\mathbf{K}^i$ means that applying permutation $\mathbf{K}^i$ to the rows of $\mathbf{W}^{i,i}$ has the same effect as applying $\mathbf{K}^i$ to the columns of $\mathbf{W}^{i,i}$. In the case of $i \neq j$, ensuring that $\mathbf{K}^i$ and $\mathbf{K}^j$ are defined over the same underlying set of permutations, $\{\mathbf{G}^d \in \mathcal{S}^{N_d} \mid d \in \mathbb{R}_i \cup \mathbb{R}_j\}$, ensures that permuting the rows of $\mathbf{W}^{i,j}$ with $\mathbf{K}^i$ has the same effect as permuting the columns of $\mathbf{W}^{i,j}$ with $\mathbf{K}^j$. It is this property that will allow us to show that a network layer defined using such a parameter tying scheme is an EERL. See Fig. 2 for a minimal example, demonstrating this lemma.

We require the following technical Lemma for the proof of Lemma 1.

**Lemma 2.** *Let* $\mathbb{R}_i, \mathbb{R}_j \in \mathfrak{R}$, *and for each* $d \in [D]$ *let* $\mathbf{G}^d \in \mathcal{S}^{N_d}$. *If* $\mathbf{G}^d_{n^i_{d_{(k)}}, n^{i\prime}_{d_{(k)}}} = 1$ *for all* $d_{(k)} \in \mathbb{R}_i$ *with* $d_{(k)} = d$, *and* $\mathbf{G}^d_{n^{j\prime}_{d_{(k)}}, n^j_{d_{(k)}}} = 1$ *for all* $d_{(k)} \in \mathbb{R}_j$ *with* $d_{(k)} = d$, *then for all* $d_{(k)} \in \mathbb{R}_i$ *and* $d_{(l)} \in \mathbb{R}_j$ *with* $d_{(k)} = d_{(l)} = d$, *we have* $n^{i\prime}_{d_{(k)}} = n^j_{d_{(l)}} \iff n^i_{d_{(k)}} = n^{j\prime}_{d_{(l)}}$.

*Proof.* Suppose $\mathbf{G}^d_{n^i_{d_{(k)}}, n^{i\prime}_{d_{(k)}}} = 1$ for all $d_{(k)} \in \mathbb{R}_i$, and $\mathbf{G}^d_{n^{j\prime}_{d_{(k)}}, n^j_{d_{(k)}}} = 1$ for all $d_{(k)} \in \mathbb{R}_j$. We prove the forward direction ($\implies$). The backward direction follows from an identical argument. Fix some $d_{(k)} \in \mathbb{R}_i$ and $d_{(l)} \in \mathbb{R}_j$ and suppose $n^{i\prime}_{d_{(k)}} = n^j_{d_{(l)}}$. By assumption we have $\mathbf{G}^d_{n^i_{d_{(k)}}, n^{i\prime}_{d_{(k)}}} = 1$ and so

$$g^d(n^{i\prime}_{d_{(k)}}) = n^i_{d_{(k)}}. \tag{31}$$

Similarly, we have $\mathbf{G}^d_{n^{j\prime}_{d_{(l)}}, n^j_{d_{(l)}}} = 1$ and so

$$g^d(n^j_{d_{(l)}}) = n^{j\prime}_{d_{(l)}}. \tag{32}$$

But $n^{i\prime}_{d_{(k)}} = n^j_{d_{(l)}}$ and substituting into (31) we have

$$g^d(n^j_{d_{(l)}}) = n^i_{d_{(k)}}. \tag{33}$$

And combining (32) and (33) gives $n^i_{d_{(k)}} = n^{j\prime}_{d_{(l)}}$, as desired.

$\square$

We are now equipped to prove our main claims, starting with Lemma 1:

*Proof.* ($\impliedby$) Let $\widetilde{\mathbb{R}_i} = \{d^i_1, ..., d^i_{|\widetilde{\mathbb{R}_i}|}\}$ and $\widetilde{\mathbb{R}_j} = \{d^j_1, ..., d^j_{|\widetilde{\mathbb{R}_j}|}\}$ and fix some $\{\mathbf{G}^d \in \mathcal{S}^{N_d} \mid d \in \mathbb{R}_i \cup \mathbb{R}_j\}$. We index the rows of $\mathbf{W}^{i,j}$, and the rows and columns of $\mathbf{K}^i$, with tuples $\mathbf{n}^i, \mathbf{n}^{i\prime} \in [N_{d^i_1}] \times ... \times [N_{d^i_{|\mathbb{R}_i|}}]$. Similarly, the columns of $\mathbf{W}^{i,j}$, and rows and columns of $\mathbf{K}^j$, are indexed with tuples $\mathbf{n}^j, \mathbf{n}^{j\prime} \in [N_{d^j_1}] \times ... \times [N_{d^j_{|\mathbb{R}_j|}}]$. Since $\mathbf{K}^i = \bigotimes_{d \in \mathbb{R}_i} \mathbf{G}^d$ we have

$$\mathbf{K}^i_{\mathbf{n}^i, \mathbf{n}^{i\prime}} = \prod_{d \in \widetilde{\mathbb{R}_i}} \mathbf{G}^d_{n^i_d, n^{i\prime}_d} = \prod_{d \in \mathbb{R}_i} \prod_{k=1}^{\kappa(d)} \mathbf{G}^d_{n^i_{d_{(k)}}, n^{i\prime}_{d_{(k)}}}.$$

And thus,

$$\mathbf{K}^i_{\mathbf{n}^i, \mathbf{n}^{i\prime}} = 1 \iff \mathbf{G}^d_{n^i_{d_{(k)}}, n^{i\prime}_{d_{(k)}}} = 1, \forall d_{(k)} \in \widetilde{\mathbb{R}_i} \text{ s.t. } d_{(k)} = d. \tag{34}$$

The same is true for $\mathbb{R}_j$. That is

$$\mathbf{K}^j_{\mathbf{n}^{j\prime}, \mathbf{n}^j} = 1 \iff \mathbf{G}^d_{n^{j\prime}_{d_{(k)}}, n^j_{d_{(k)}}} = 1, \forall d_{(k)} \in \widetilde{\mathbb{R}_j} \text{ s.t. } d_{(k)} = d. \tag{35}$$

Now, fix some $\mathbf{n}^i$ and $\mathbf{n}^j$. Since $\mathbf{K}^i$ is a permutation matrix, and so has only one 1 per row, we have

$$
\begin{aligned}
\left(\mathbf{K}^i\mathbf{W}^{i,j}\right)_{\mathbf{n}^i,\mathbf{n}^j} &= \sum_{\mathbf{n}^{i\prime}} \mathbf{K}^i_{\mathbf{n}^i,\mathbf{n}^{i\prime}}\, \mathbf{W}^{i,j}_{\mathbf{n}^{i\prime}\mathbf{n}^j} \\
&= \mathbf{W}^{i,j}_{\mathbf{n}^{i*},\mathbf{n}^j},
\end{aligned}
\tag{36}
$$

where $\mathbf{n}^{i*}$ is the (unique) element of $[N_{d_1^i}] \times ... \times [N_{d_{|R_i|}^i}]$ which satisfies $\mathbf{G}^d_{n^i_{d_{(k)}},n^{i*}_{d_{(k)}}} = 1$ for all $d_{(k)} \in \widetilde{\mathbb{R}}_{\mathbb{i}}$ with $d_{(k)} = d$. Similarly,

$$
\begin{aligned}
\left(\mathbf{W}^{i,j}\mathbf{K}^j\right)_{\mathbf{n}^i,\mathbf{n}^j} &= \sum_{\mathbf{n}^{j\prime}} \mathbf{W}^{i,j}_{\mathbf{n}^i,\mathbf{n}^{j\prime}}\, \mathbf{K}^j_{\mathbf{n}^{j\prime},\mathbf{n}^j} \\
&= \mathbf{W}^{i,j}_{\mathbf{n}^i,\mathbf{n}^{j*}}
\end{aligned}
\tag{37}
$$

where $\mathbf{n}^{j*}$ is the (unique) element of $[N_{d_1^j}] \times ... \times [N_{d_{|\mathbb{R}_j|}^j}]$ which satisfies $\mathbf{G}^d_{n^{j*}_{d_{(k)}},n^j_{d_{(k)}}} = 1$ for all $d_{(k)} \in \widetilde{\mathbb{R}}_{\mathbb{j}}$ with $d_{(k)} = d$.

We want to show that $\mathbb{P}(\mathbf{n}_d^{i^*,j}) = \mathbb{P}(\mathbf{n}_d^{i,j^*})$ for all $d \in \mathbb{R}_i \cup \mathbb{R}_j$. Fix $d \in \mathbb{R}_i \cup \mathbb{R}_j$ and let $\widetilde{\mathbb{S}} \in \mathbb{P}(\mathbf{n}_d^{i^*,j})$. Then $\widetilde{\mathbb{S}} = \{d_{(1)}, ..., d_{(K)}\}$, where $d_{(k)} = d$ for all $d_{(k)} \in \widetilde{\mathbb{S}}$, and for all $d_{(k)}, d_{(l)} \in \widetilde{\mathbb{S}}$, $n_{d_{(k)}}^{i^*,j} = n_{d_{(l)}}^{i^*,j}$. Then by Lemma 2 we have $n_{d_{(k)}}^{i,j^*} = n_{d_{(l)}}^{i,j^*}$, and so $\widetilde{\mathbb{S}} \in \mathbb{P}(\mathbf{n}_d^{i,j^*})$. So we have $\mathbb{P}(\mathbf{n}_d^{i^*,j}) \subseteq \mathbb{P}(\mathbf{n}_d^{i,j^*})$, and the other containment follows identically by symmetry. So $\mathbf{n}^{i^*,j} \equiv \mathbf{n}^{i,j^*}$ by our definition in Section 4.1, and so $\mathbf{W}^{i,j}_{\mathbf{n}^{i*},\mathbf{n}^j} = \mathbf{W}^{i,j}_{\mathbf{n}^i,\mathbf{n}^{j*}}$ and by (36) and (37) above, $\mathbf{K}^i\mathbf{W}^{i,j} = \mathbf{W}^{i,j}\mathbf{K}^j$.

($\Longrightarrow$) Suppose $\mathbf{K}^i\mathbf{W}^{i,j} = \mathbf{W}^{i,j}\mathbf{K}^j$. Fix some $\mathbf{n}^i, \mathbf{n}^j$. Let $\mathbf{n}^{i^*}$ be the unique index such that $\mathbf{K}^i_{\mathbf{n}^i,\mathbf{n}^{i*}} = 1$, and $\mathbf{n}^j$ the unique index such that $\mathbf{K}^j_{\mathbf{n}^{j*},\mathbf{n}^j} = 1$. Then

$$
\begin{aligned}
\mathbf{W}^{i,j}_{\mathbf{n}^{i*},\mathbf{n}^j} &= \sum_{\mathbf{n}^{i\prime}} \mathbf{K}^i_{\mathbf{n}^i,\mathbf{n}^{i\prime}}\mathbf{W}^{i,j}_{\mathbf{n}^{i\prime},\mathbf{n}^j} \\
&= \left(\mathbf{K}^i\mathbf{W}^{i,j}\right)_{\mathbf{n}^i,\mathbf{n}^j} \\
&= \left(\mathbf{W}^{i,j}\mathbf{K}^j\right)_{\mathbf{n}^i,\mathbf{n}^j} \\
&= \sum_{\mathbf{n}^{j\prime}} \mathbf{W}^{i,j}_{\mathbf{n}^i,\mathbf{n}^{j\prime}}\mathbf{K}^j_{\mathbf{n}^{j\prime},\mathbf{n}^j} \\
&= \mathbf{W}^{i,j}_{\mathbf{n}^i,\mathbf{n}^{j*}},
\end{aligned}
\tag{38}
$$

and so $\mathbb{P}(\mathbf{n}_d^{i^*,j}) = \mathbb{P}(\mathbf{n}_d^{i,j^*})$ for all $d \in \mathbb{R}_i \cup \mathbb{R}_j$. But this implies that

$$
\begin{aligned}
\mathbb{P}(\mathbf{n}_d^{i^*}) &= \mathbb{P}(\mathbf{n}_d^i) \\
&= \mathbb{P}\left((k^i(\mathbf{n}^{i^*}))_d\right).
\end{aligned}
\tag{39}
$$

(39) says that for each $d$ the action of $\mathbf{K}^i$ on elements of $\mathbf{n}^{i^*}$ is determined only by the values of those elements, not by the values of elements indexing other entities, and so $\mathbf{K}^i \in \bigotimes_{d \in \mathbb{R}_i} \mathcal{S}^{N_d}$. But (39) also means that for all $k, l$

$$
n_{d_{(k)}}^{i^*} = n_{d_{(l)}}^{i^*} \iff \left(k^i(\mathbf{n}^{i^*})\right)_{d_{(k)}} = \left(k^i(\mathbf{n}^{i^*})\right)_{d_{(l)}},
\tag{40}
$$

which says that the action of $\mathbf{K}^i$ is the same across any duplications of $d$ (i.e., $d_{(k)}$ and $d_{(l)}$), and so $\mathbf{K}^i = \bigotimes_{d \in \mathbb{R}_i} \mathbf{G}^d$, for some fixed $\mathbf{G}^d$. Similarly,

$$
\begin{aligned}
\mathbb{P}(\mathbf{n}_d^j) &= \mathbb{P}(\mathbf{n}_d^{j^*}) \\
&= \mathbb{P}\left((k^j(\mathbf{n}^j))_d\right),
\end{aligned}
\tag{41}
\tag{42}
$$

which shows that $\mathbf{K}^j = \bigotimes_{d \in \mathbb{R}_j} \mathbf{G}^{N_d\prime}$. Finally, since $\mathbb{P}(\mathbf{n}_d^{i^*,j}) = \mathbb{P}(\mathbf{n}_d^{i,j^*})$, we also have

$$
n_{d_{(k)}}^{i^*} = n_{d_{(l)}}^j \iff n_{d_{(k)}}^i = n_{d_{(l)}}^{j^*},
\tag{43}
$$

for all $k, l$, which means

$$n_{d_{(k)}}^{i^*} = n_{d_{(l)}}^j \iff \left(k^i(\mathbf{n}^{i^*})\right)_{d_{(k)}} = \left(k^j(\mathbf{n}^j)\right)_{d_{(l)}}. \tag{44}$$

(44) says that $\mathbf{K}^i$ and $\mathbf{K}^j$ apply the same permutations to all duplications of any entities they have in common, and so $\mathbf{G}^d = \mathbf{G}^{d'}$, which completes the proof. $\qquad\square$

### G.3 PROOF OF THEOREM 4.1

*Proof.* Let $\mathbf{G}^{\mathbb{X}} \in \mathcal{S}^N$ and $\mathcal{G}^{\mathbb{X}}$ be defined as in Claim 1. We need to show that $\mathbf{G}^{\mathbb{X}}\sigma(\mathbf{W}\,\mathrm{vec}(\mathbb{X})) = \sigma(\mathbf{W}\mathbf{G}^{\mathbb{X}}\,\mathrm{vec}(\mathbb{X}))$ iff $\mathbf{G}^{\mathbb{X}} \in \mathcal{G}^{\mathbb{X}}$ for any assignment of values to the tables $\mathbb{X}$. We prove each direction in turn.

($\Longrightarrow$) We prove the contrapositive. Suppose $\mathbf{G}^{\mathbb{X}} \notin \mathcal{G}^{\mathbb{X}}$. We first show that $\mathbf{G}^{\mathbb{X}}\mathbf{W} \neq \mathbf{W}\mathbf{G}^{\mathbb{X}}$ and then that $\mathbf{G}^{\mathbb{X}}\sigma(\mathbf{W}\,\mathrm{vec}(\mathbb{X})) \neq \sigma(\mathbf{W}\mathbf{G}^{\mathbb{X}}\,\mathrm{vec}(\mathbb{X}))$ for an appropriate choice of $\mathbb{X}$. There are three cases. First, suppose $\mathbf{G}^{\mathbb{X}} \notin \bigoplus_{\mathbb{R} \in \mathfrak{R}} \mathcal{S}^{N_{\mathbb{R}}}$ and consider dividing the rows and columns of $\mathbf{G}^{\mathbb{X}}$ into blocks that correspond to the blocks of $\mathbf{W}$. Then since $\mathbf{G}^{\mathbb{X}} \notin \bigoplus_{\mathbb{R} \in \mathfrak{R}} \mathcal{S}^{N_{\mathbb{R}}}$, there exist $\mathbb{R}_i, \mathbb{R}_j \in \mathfrak{R}$, with $i \neq j$ and $\mathbf{n}^i \in [N_{d_1^i}] \times ... \times [N_{d_{|\mathbb{R}_i|}^i}]$ and $\mathbf{n}^j \in [N_{d_1^j}] \times ... \times [N_{d_{|\mathbb{R}_j|}^j}]$ such that $\mathbf{G}^{\mathbb{X}}$ maps $(i, \mathbf{n}^i)$ to $(j, \mathbf{n}^j)$. That is $g^{\mathbb{X}}\left((i, \mathbf{n}^i)\right) = (j, \mathbf{n}^j)$ and thus, $\mathbf{G}^{\mathbb{X}}_{(j,\mathbf{n}^j),(i,\mathbf{n}^i)} = 1$. And so

$$\left(\mathbf{G}^{\mathbb{X}}\mathbf{W}\right)_{(j,\mathbf{n}^j),(i,\mathbf{n}^i)} = \sum_k \sum_{\mathbf{n}^k} \mathbf{G}^{\mathbb{X}}_{(j,\mathbf{n}^j),(k,\mathbf{n}^k)} \mathbf{W}_{(k,\mathbf{n}^k),(i,\mathbf{n}^i)}$$
$$= \mathbf{W}^{i,i}_{\mathbf{n}^i,\mathbf{n}^i},$$

by the definition of $\mathbf{W}$. Similarly,

$$\left(\mathbf{W}\mathbf{G}^{\mathbb{X}}\right)_{(j,\mathbf{n}^j),(i,\mathbf{n}^i)} = \sum_k \sum_{\mathbf{n}^k} \mathbf{W}_{(j,\mathbf{n}^j),(k,\mathbf{n}^k)} \mathbf{G}^{\mathbb{X}}_{(k,\mathbf{n}^k),(i,\mathbf{n}^i)}$$
$$= \mathbf{W}^{j,j}_{\mathbf{n}^j,\mathbf{n}^j}$$

But $\mathbf{W}^{i,i}_{\mathbf{n}^i,\mathbf{n}^i} \neq \mathbf{W}^{j,j}_{\mathbf{n}^j,\mathbf{n}^j}$ since $i \neq j$. And so $\mathbf{G}^{\mathbb{X}}\mathbf{W} \neq \mathbf{W}\mathbf{G}^{\mathbb{X}}$.

Next, suppose $\mathbf{G}^{\mathbb{X}} \in \bigoplus_{\mathbb{R} \in \mathfrak{R}} \mathcal{S}^{N_{\mathbb{R}}}$, but $\mathbf{G}^{\mathbb{X}} \notin \bigoplus_{\mathbb{R} \in \mathfrak{R}} \bigotimes_{d \in \mathbb{R}} \mathcal{S}^{N_d}$ and consider the diagonal blocks of $\mathbf{G}^{\mathbb{X}}\mathbf{W}\mathbf{G}^{\mathbb{X}^T}$ that correspond to those of $\mathbf{W}$. If $\mathbf{G}^{\mathbb{X}} \in \bigoplus_{\mathbb{R} \in \mathfrak{R}} \mathcal{S}^{N_R}$ then it is block diagonal with blocks corresponding to each $\mathbb{R} \in \mathfrak{R}$. But since $\mathbf{G}^{\mathbb{X}} \notin \bigoplus_{\mathbb{R} \in \mathfrak{R}} \bigotimes_{d \in \mathbb{R}} \mathcal{S}^{N_d}$, there exists some $\mathbb{R}_i \in \mathfrak{R}$ such that the $i^{\text{th}}$ diagonal block of $\mathbf{G}^{\mathbb{X}}$ is not of the form $\bigotimes_{d \in \mathbb{R}_i} \mathbf{G}^d$ for any $\mathbf{G}^d$. Then by Lemma 1 we will have inequality between $\mathbf{G}^{\mathbb{X}}\mathbf{W}\mathbf{G}^{\mathbb{X}^T}$ and $\mathbf{W}$ in the $i^{\text{th}}$ diagonal block.

Finally, suppose $\mathbf{G}^{\mathbb{X}} \in \bigoplus_{\mathbb{R} \in \mathfrak{R}} \bigotimes_{d \in \mathbb{R}} \mathcal{S}^{N_d}$. Then $\mathbf{G}^{\mathbb{X}} = \mathbf{K}^1 \oplus ... \oplus \mathbf{K}^{|\mathfrak{R}|}$, where $\mathbf{K}^i \in \bigotimes_{d \in \mathbb{R}} \mathcal{S}^{N_d}$ for all $i$. Since $\mathbf{G}^{\mathbb{X}} \notin \mathcal{G}^{\mathbb{X}}$, there exist $\widetilde{\mathbb{R}_i}, \widetilde{\mathbb{R}_j} \in \mathfrak{R}$, possibly with $i = j$, and a $d^* \in \mathbb{R}_i \cap \mathbb{R}_j$ such that

$$\mathbf{K}^i = \mathbf{G}^{d_1^i} \otimes ... \otimes \mathbf{G}^{d_{(k)}^*} \otimes ... \otimes \mathbf{G}^{d_{|\mathbb{R}_i|}^i},$$

and

$$\mathbf{K}^j = \mathbf{G}^{d_1^j} \otimes ... \otimes \mathbf{G}^{d_{(l)}^*} \otimes ... \otimes \mathbf{G}^{d_{|\mathbb{R}_j|}^j},$$

but $\mathbf{G}^{d_{(k)}^*} \neq \mathbf{G}^{d_{(l)}^*}$. Since $\mathbf{G}^{d_{(k)}^*} \neq \mathbf{G}^{d_{(l)}^*}$ there exists $n \in [N_{d^*}]$ with $g^{d_{(l)}^*}(n) \neq g^{d_{(l)}^*}(n)$. Pick some $\mathbf{n}^i$ and $\mathbf{n}^j$ with $n_{d^*}^i = n_{d^*}^j = n$. Let $\mathbf{n}^{i^*}$ be the result of applying $\mathbf{K}^i$ to $\mathbf{n}^i$ and $\mathbf{n}^{j^*}$ the result of applying $\mathbf{K}^j$ to $\mathbf{n}^j$. Then we have

$$\left(\mathbf{G}^{\mathbb{X}}\mathbf{W}\right)_{(i,\mathbf{n}^{i^*}),(j,\mathbf{n}^j)} = \sum_k \sum_{\mathbf{n}^k} \mathbf{G}^{\mathbb{X}}_{(i,\mathbf{n}^{i^*}),(k,\mathbf{n}^k)} \mathbf{W}_{(k,\mathbf{n}^k),(j,\mathbf{n}^j)}$$
$$= \sum_{\mathbf{n}^k} \mathbf{K}^i_{\mathbf{n}^{i^*},\mathbf{n}^k} \mathbf{W}^{i,j}_{\mathbf{n}^k,\mathbf{n}^j}$$
$$= \mathbf{W}^{i,j}_{\mathbf{n}^i,\mathbf{n}^j}, \tag{45}$$

and

$$
\begin{aligned}
\left(\mathbf{W}\mathbf{G}^{\mathbb{X}}\right)_{(i,\mathbf{n}^{i*}),(j,\mathbf{n}^{j})} &= \sum_{k}\sum_{\mathbf{n}^{k}} \mathbf{W}_{(i,\mathbf{n}^{i*}),(k,\mathbf{n}^{k})} \mathbf{G}^{\mathbb{X}}_{(k,\mathbf{n}^{k}),(j,\mathbf{n}^{j})} \\
&= \sum_{\mathbf{n}^{k}} \mathbf{W}^{i,j}_{\mathbf{n}^{i*},\mathbf{n}^{k}} \mathbf{K}^{j}_{\mathbf{n}^{k},\mathbf{n}^{j}} \\
&= \mathbf{W}^{i,j}_{\mathbf{n}^{i*},\mathbf{n}^{j*}}.
\end{aligned}
\tag{46}
$$

Now, by construction we have $n^{i}_{d*} = n^{j}_{d*}$, but $n^{i^{*}}_{d*} \neq n^{j^{*}}_{d*}$. So $\mathbb{P}(\mathbf{n}^{i,j}_{d*}) \neq \mathbb{P}(\mathbf{n}^{i^{*},j^{*}}_{d*})$ and therefore $\mathbf{W}^{i,j}_{\mathbf{n}^{i},\mathbf{n}^{j}} \neq \mathbf{W}^{i,j}_{\mathbf{n}^{i*},\mathbf{n}^{j*}}$. And so by (45) and (46) we have $\mathbf{G}^{\mathbb{X}}\mathbf{W} \neq \mathbf{W}\mathbf{G}^{\mathbb{X}}$.

And so in all three cases $\mathbf{G}^{\mathbb{X}}\mathbf{W} \neq \mathbf{W}\mathbf{G}^{\mathbb{X}}$. Thus, there exists some $\mathbb{X}$, for which we have $\mathbf{G}^{\mathbb{X}}\mathbf{W}\,\mathrm{vec}(\mathbb{X}) \neq \mathbf{W}\mathbf{G}^{\mathbb{X}}\,\mathrm{vec}(\mathbb{X})$. Since $\sigma$ is strictly monotonic, we have $\sigma(\mathbf{G}^{\mathbb{X}}\mathbf{W}\,\mathrm{vec}(\mathbb{X})) \neq \sigma(\mathbf{W}\mathbf{G}^{\mathbb{X}}\,\mathrm{vec}(\mathbb{X}))$. And since $\sigma$ is element wise we have $\mathbf{G}^{\mathbb{X}}\sigma(\mathbf{W}\,\mathrm{vec}(\mathbb{X})) \neq \sigma(\mathbf{W}\mathbf{G}^{\mathbb{X}}\,\mathrm{vec}(\mathbb{X}))$, which proves the first direction.

($\Longleftarrow$) Suppose $\mathbf{G}^{\mathbb{X}} \in \mathcal{G}^{\mathbb{X}}$. That is, for all $d \in [D]$, let $\mathbf{G}^{d} \in \mathcal{S}^{N_{d}}$ be some fixed permutation of $N_{d}$ objects and let $\mathbf{G}^{\mathbb{X}} = \bigoplus_{\mathbb{R}\in\mathfrak{R}} \bigotimes_{d\in\mathbb{R}} \mathbf{G}^{d}$. Observe that $\mathbf{G}^{\mathbb{X}}$ is block-diagonal. Each block on the diagonal corresponds to an $\mathbb{R} \in \mathfrak{R}$ and is a Kronecker product over the matrices $\mathbf{G}^{d}$ for each $d \in \mathbb{R}$. Let $\mathbf{K}^{i} = \bigotimes_{d\in\mathbb{R}_{i}} \mathbf{G}^{d}$ for each $i \in [|\mathfrak{R}|]$. That is,

$$
\mathbf{G}^{\mathbb{X}} = \begin{bmatrix} \mathbf{K}^{1} & & & \mathbf{0} \\ & \mathbf{K}^{2} & & \\ & & \ddots & \\ \mathbf{0} & & & \mathbf{K}^{|\mathfrak{R}|} \end{bmatrix}.
$$

And so the $i,j$-th block of $\mathbf{G}^{\mathbb{X}}\mathbf{W}\mathbf{G}^{\mathbb{X}^{T}}$ is given by:

$$
\begin{aligned}
\left(\mathbf{G}^{\mathbb{X}}\mathbf{W}\mathbf{G}^{\mathbb{X}^{T}}\right)^{i,j} &= \mathbf{K}^{i}\mathbf{W}^{i,j}\mathbf{K}^{j^{T}} \\
&= \mathbf{W}^{i,j}.
\end{aligned}
\tag{47}
$$

The equality at (47) follows from Lemma 1. Thus, we have $\mathbf{G}^{\mathbb{X}}\mathbf{W} = \mathbf{W}\mathbf{G}^{\mathbb{X}}$, and so for all $\mathbb{X}$, $\mathbf{G}^{\mathbb{X}}\mathbf{W}\,\mathrm{vec}(\mathbb{X}) = \mathbf{W}\mathbf{G}^{\mathbb{X}}\,\mathrm{vec}(\mathbb{X})$. Finally, since $\sigma$ is applied element-wise, we have

$$
\begin{aligned}
\sigma(\mathbf{W}\mathbf{G}^{\mathbb{X}}\,\mathrm{vec}(\mathbb{X})) &= \sigma(\mathbf{G}^{\mathbb{X}}\mathbf{W}\,\mathrm{vec}(\mathbb{X})) \\
&= \mathbf{G}^{\mathbb{X}}\sigma(\mathbf{W}\,\mathrm{vec}(\mathbb{X}))
\end{aligned}
$$

Which proves the second direction. And so $\sigma(\mathbf{W}\,\mathrm{vec}(\mathbb{X}))$ is an exchangeable relation layer, completing the proof. $\qquad\square$

### G.4 PROOF OF THEOREM 4.2

The idea is that if $\mathbf{W}$ is not of the form (4) then it has some block $\mathbf{W}^{i,j}$ containing two elements whose indices have the same equality pattern, but whose values are different. Based on these indices, we can explicitly construct a permutation which swaps the corresponding elements of these indices. This permutation is in $\mathcal{G}^{\mathbb{X}}$ but it does not commute with $\mathbf{W}$. Now we present a detailed proof.

*Proof.* Let $\mathbf{G}^{\mathbb{X}} \in \mathcal{S}^{N}$. For any relation $\mathbb{R} = \{d_{1}, ..., d_{|\mathbb{R}|}\} \in \mathfrak{R}$, let $\mathbb{N}^{\mathbb{R}} = [N_{d_{1}}] \times ... \times [N_{d_{|\mathbb{R}|}}]$ be the set of indices into $\mathbb{X}^{\mathbb{R}}$. If $\mathbf{W}$ is not of the form described in Section 4 then there exist $i, j \in [|\mathfrak{R}|]$, with $\mathbf{n}^{i}, \mathbf{n}^{i'} \in \mathbb{N}^{\mathbb{R}_{i}}$ and $\mathbf{n}^{j}, \mathbf{n}^{j'} \in \mathbb{N}^{\mathbb{R}_{j}}$ such that

$$
\mathbf{W}^{i,j}_{\mathbf{n}^{i},\mathbf{n}^{j}} \neq \mathbf{W}^{i,j}_{\mathbf{n}^{i'},\mathbf{n}^{j'}}
\tag{48}
$$

but

$$
\mathbb{P}(\mathbf{n}^{i,j}_{d}) = \mathbb{P}(\mathbf{n}^{i',j'}_{d}), \ \forall d \in \mathbb{R}_{i} \cup \mathbb{R}_{j}
\tag{49}
$$

That is, the pairs $\mathbf{n}^i$, $\mathbf{n}^j$ and $\mathbf{n}^{i'}$, $\mathbf{n}^{j'}$ have the same equality pattern over their elements, but the entries of $\mathbf{W}^{i,j}$ which correspond to these pairs have differing values, and thus violate the definition of $\mathbf{W}$ in Section 4. To show that the layer $\sigma(\mathbf{W}\operatorname{vec}(\mathbb{X}))$ is not an EERL, we will demonstrate a permutation $\mathbf{G}^{\mathbb{X}} \in \mathcal{G}^{\mathbb{X}}$ for which $\mathbf{G}^{\mathbb{X}}\mathbf{W} \neq \mathbf{W}\mathbf{G}^{\mathbb{X}}$, and thus $\mathbf{G}^{\mathbb{X}}\sigma(\mathbf{W}\operatorname{vec}(\mathbb{X})) \neq \sigma(\mathbf{W}\mathbf{G}^{\mathbb{X}}\operatorname{vec}(\mathbb{X}))$ for some $\mathbb{X}$.

Let $\mathbf{G}^{\mathbb{X}} = \bigoplus_{\mathbb{R} \in \mathfrak{R}} \bigotimes_{d \in \mathbb{R}} \mathbf{G}^d$, with the $\mathbf{G}^d$ defined as follows. For $d \in \mathbb{R}_i \cap \mathbb{R}_j$ and all $k$:

$$\mathbf{G}^d(n) = \begin{cases} n^{i'}_{d_{(k)}} & n = n^i_{d_{(k)}} \\ n^i_{d_{(k)}} & n = n^{i'}_{d_{(k)}} \\ n^{j'}_{d_{(k)}} & n = n^j_{d_{(k)}} \\ n^j_{d_{(k)}} & n = n^{j'}_{d_{(k)}} \\ n & \text{otherwise} \end{cases}.$$

For $d \in \mathbb{R}_i \backslash \mathbb{R}_j$ and all $k$:

$$\mathbf{G}^d(n) = \begin{cases} n^{i'}_{d_{(k)}} & n = n^i_{d_{(k)}} \\ n^i_{d_{(k)}} & n = n^{i'}_{d_{(k)}} \\ n & \text{otherwise} \end{cases}.$$

For $d \in \mathbb{R}_j \backslash \mathbb{R}_i$ and all $k$:

$$\mathbf{G}^d(n) = \begin{cases} n^{j'}_{d_{(k)}} & n = n^j_{d_{(k)}} \\ n^j_{d_{(k)}} & n = n^{j'}_{d_{(k)}} \\ n & \text{otherwise} \end{cases}.$$

And for $d \notin \mathbb{R}_i \cup \mathbb{R}_j$:

$$\mathbf{G}^d(n) = n.$$

That is, each $\mathbf{G}^d$ swaps the elements of $\mathbf{n}^i$ with the corresponding elements of $\mathbf{n}^{i'}$, and the elements of $\mathbf{n}^j$ with those of $\mathbf{n}^{j'}$, so long as the relevant indices are present. For the case where $d \in \mathbb{R}_i \cap \mathbb{R}_j$, we need to make sure that this is a valid permutation. Specifically, we need to make sure that it is injective (it is clearly surjective from $[N_d]$ to $[N_d]$). But it is indeed injective, since we have $n^i_{d_{(k)}} = n^j_{d_{(k)}}$ iff $n^{i'}_{d_{(k)}} = n^{j'}_{d_{(k)}}$ for all $d \in \mathbb{R}_i \cap \mathbb{R}_j$ and all $k$, since $\mathbb{P}(\mathbf{n}^{i,j}_d) = \mathbb{P}(\mathbf{n}^{i',j'}_d)$.

Now, for all $i$, let $\mathbf{K}^i = \bigotimes_{d \in \mathbb{R}_i} \mathbf{G}^d$ be the $i^{\text{th}}$ diagonal block of $\mathbf{G}^{\mathbb{X}}$. By definition of the $\mathbf{G}^d$, for all $d \in \mathbb{R}_i$ we have $\mathbf{G}^d_{n^i_d, n^{i'}_d} = 1$, and thus by the observation at (34) we have $\mathbf{K}^i_{\mathbf{n}^i, \mathbf{n}^{i'}} = 1$. And so

$$\left(\mathbf{G}^{\mathbb{X}}\mathbf{W}\right)_{(i,\mathbf{n}^i),(j,\mathbf{n}^{j'})} = \sum_k \sum_{\mathbf{n}^k} \mathbf{G}^{\mathbb{X}}_{(i,\mathbf{n}^i),(k,\mathbf{n}^k)} \mathbf{W}_{(k,\mathbf{n}^k),(j,\mathbf{n}^{j'})}$$
$$= \sum_{\mathbf{n}^{i''}} \mathbf{K}^i_{\mathbf{n}^i, \mathbf{n}^{i''}} \mathbf{W}^{i,j}_{\mathbf{n}^{i''}, \mathbf{n}^{j'}}$$
$$= \mathbf{W}^{i,j}_{\mathbf{n}^{i'}, \mathbf{n}^{j'}}.$$

Similarly, $\mathbf{K}^j_{\mathbf{n}^j, \mathbf{n}^{j'}} = 1$, so

$$\left(\mathbf{W}\mathbf{G}^{\mathbb{X}}\right)_{(i,\mathbf{n}^i),(j,\mathbf{n}^{j'})} = \sum_k \sum_{\mathbf{n}^k} \mathbf{W}_{(i,\mathbf{n}^i),(k,\mathbf{n}^k)} \mathbf{G}^{\mathbb{X}}_{(k,\mathbf{n}^k),(j,\mathbf{n}^{j'})}$$
$$= \sum_{\mathbf{n}^{j''}} \mathbf{W}^{i,j}_{\mathbf{n}^i, \mathbf{n}^{j''}} \mathbf{K}^j_{\mathbf{n}^{j''}, \mathbf{n}^{j'}}$$
$$= \mathbf{W}^{i,j}_{\mathbf{n}^i, \mathbf{n}^j}.$$

Finally, by (48), $\mathbf{G}^{\mathbb{X}}\mathbf{W} \neq \mathbf{W}\mathbf{G}^{\mathbb{X}}$, completing the proof.

$\square$

## H  DETAILS OF EXPERIMENTS

The synthetic data we constructed used 200 instances for each of the three entities. Unless otherwise noted, we considered a 0.1 fraction as 'observed'. We ensure that each row and column has at least 5 observations. For training, we pass our observed data through the network, producing encodings for each entity. We attempt to reconstruct the original input from these encodings. At test time, we use the training data to produce encodings as before, but now use these encodings to attempt to reconstruct test observations as well, reporting loss only on these test observations.

For all experiments, we used the following architecture: The encoder consists of 7 EERLs, each with 64 hidden units, each followed by a batch normalization (Ioffe & Szegedy, 2015) layer and a channel-wise dropout layer. We then apply a mean pooling layer to produce encodings. We found that batch normalization dramatically sped up the training procedure. The number of units in the pooling layer, and hence the size of the encodings, varied by experiment (see below for details). The decoder architecture is identical to that of the encoder, except that there is no pooling layer before the output. We used leaky ReLU activations (with parameter 0.1) in all layers except the final ones in both the encoder and decoder, where no activation is applied. In all experiments we used the Adam (Kingma & Ba, 2014) optimizer with initial learning rate 0.0001 to optimize a loss between the observed input values and our predicted output values in the relevant table(s). FOr the synthetic experiments the loss was RMSE, while for the soccer experiment we used cross-entropy. We did not perform any tuning of hyperparameters.

### H.1  EMBEDDING

We ran this experiment for 5000 epochs, and used an encoding size of 2 to facilitate plotting. We trained the model on 90% of the data, using 80% for training and 10% for validation. The remaining 10% was used to evaluate the quality of the model learned.

### H.2  MISSING RECORD PREDICTION

We held out a special 10% portion of the data for testing, then trained models using various amounts of the remaining data. For a 90%, 80%, ..., 10% fraction of the remaining data, we trained a model using this fraction of the data as observed (using a 90%/10% training/validation split). We then used each of these trained models to predict the held-out test set. When predicting the test set, we again varied the fraction of observed data we gave to the model to make its predictions, from 90% to 10%. For the inductive setting, the trained models were given data from the same distribution as they were trained on, while in the inductive setting the models were given data generated from a new set of entity embeddings. For each sparsity level, we trained the model for 4000 epochs, using an encoding size of 10. We repeated this experiment 15 times and averaged the results.

### H.3  PREDICTIVE VALUE OF SIDE INFORMATION

We fix the sparsity level of the STUDENT-COURSE table at 10% and calculate loss only on this table. We vary the sparsity level of the other tables over the range [.025, .7]. We use an 80%/10%/10% training/validation/test split and trained for 8000 epochs, producing an embedding of size 10. We repeat this experiment 17 times and average the results.

### H.4  PREDICTIONS ON THE SOCCER DATABASE

The dataset is available at https://www.kaggle.com/hugomathien/soccer. We constructed two tables for this experiment. The first table represents the relation between TEAMS and MATCHES. For input features we used all the player position features, as well as the match's country, season, stage and date. We did not include any of the bookie predictions or the number of goals, as this would be cheating. For each team, we include all the team-related features found in the Team_Attributes table. When a team appears multiple times in the Team_Attributes table we aggregate the entries by averaging numerical features, and taking the most common value for categorical ones. We also include an additional binary feature indicating which team is home and which is away.

The second table captures the relation between PLAYERS and MATCHES. For each player participating in a match, we include all the features in the Player_Attributes table, aggregating entries for each player as described above. Again, we also include a binary feature indicating whether a player played for the home or away team.

We use a model built from five EERLs. All but the last one have 52 units (this is based on the limitation of our GPU memory), use ReLU activations, and are followed by a batch normalization layer (Ioffe & Szegedy, 2015) and channel-dropout. Chennel-dropout zeros whole channels at random for all entries of all tables, at a rate of 50%. The final EERL uses no activation and 8 output units. This is followed by a layer which sum-pools both tables into a single feature vector for each match. Finally, a linear layer produces a length-3 vector of predictions for each match. We train by minimizing cross-entropy loss between this output and the true match outcomes for 4000 steps. We report test results on a $90\%/10\%$ training/test split.

