# OpenReview forum: "Equivariant Entity-Relationship Networks"
_ICLR.cc/2020/Conference — Reject_

### Official Review · AnonReviewer2 · 2019-10-22
**Official Blind Review #2**

**Rating:** 3

**Review:**

The paper handles the permutation invariance of the entity-relationship data entries by tying the weights of linear parameterization together. Heavy math was used to describe the proposed idea, and experiments were performed on simple problems.

The main idea of the paper seems to be equation (3) and the associated method of tying weights in the linear layer. This is actually quite simple, and does not require the numerous complicated definitions in section 2 and 3 to reach the same conclusion. Using math is good, but using more than necessary simply slows down the dissemination of ideas.

In both theory and experiment, the paper does not demonstrate the advantage of the weight tying method compared to an untied baseline. This weakens its necessity and motivation.

This paper also has very weak relationship with deep neural networks, while the word 'deep learning' was mentioned in both introduction and conclusion.

In general, it is a weakly motivated and math-excessive paper. Based on these considerations, I recommend rejection.

**Experience Assessment:**

I do not know much about this area.

**Review Assessment: Checking Correctness Of Derivations And Theory:**

I assessed the sensibility of the derivations and theory.

**Review Assessment: Checking Correctness Of Experiments:**

I assessed the sensibility of the experiments.

**Review Assessment: Thoroughness In Paper Reading:**

I made a quick assessment of this paper.

---

> ### Author Response · Authors · 2019-11-07
> **RE: review #2**
>
> Thanks for your review.
>
> REVIEW: " The main idea of the paper seems to be equation (3) and ... This is actually quite simple, and does not require the numerous complicated definitions in section 2 and 3 to reach the same conclusion..."
>
> RESPONSE: Equation (3) is simply the definition of equivariance, as it applies to a feedforward layer. Equation (3) on its own does not say anything about the underlying structure (it applies to an input that may be an image, a set, a graph, or a relational database). One novel contribution of this paper is the identification of the symmetry group for the relational model. Sections 2 and 3 formalize the representation and derive this symmetry group.
>
> REVIEW: "In both theory and experiment, the paper does not demonstrate the advantage of the weight tying method compared to an untied baseline..."
>
> RESPONSE: The number of parameters in a fully connected layer applied to a relational database grows quadratically with the size of the database. Even if we could construct this for a small toy problem, such a model will have a very poor generalization. That might be why fully connected models have not been used as a baseline in the related equivariant models (e.g., see Hartford et al’18 or Maron et al’18). In theory, it is known that in terms of sample complexity, an equivariant layer is comparable to a fully connected layer that uses data-augmentation via the symmetry group. In this case, due to the exponential size of the group, such data-augmentation is infeasible, further motivating our model. We will add this explanation to the revised version.
>
> REVIEW: "This paper also has very weak relationship with deep neural networks … "
>
> RESPONSE: Could you elaborate? This paper is about building deep neural networks for relational data.

---

> > ### Comment · AnonReviewer2 · 2019-11-13
> > **Deep network connection**
> >
> > Thanks to the authors to provide the response!
> >
> > For the relationship with deep neural network, could you provide pointers to the paper in which the weight tying is applied to more than a first layer of a multi-layer model with non-linearity in between?

---

> > > ### Author Response · Authors · 2019-11-13
> > > **RE: deep network connection**
> > >
> > > Thank you for the follow-up.
> > >
> > > The models that we use all have several parameter-sharing layers, followed by nonlinearity. For the autoencoding architecture, we have 7 layers in the encoder and decoder each. For the real-world experiment, we use 5 layers. Details of the architecture are in Appendix H. In the revised version, we have moved some of this information to the main body.

---

> > > > ### Comment · AnonReviewer2 · 2019-11-13
> > > > **RE**
> > > >
> > > > Thank you for the quick response! I read the paper again, and have the following considerations:
> > > >
> > > > 1. The response correctly pointed out that weight tying does help in reducing the number of parameters in a linear layer, which could be useful. The paper should stress this more, and even better, provides learning-theory style bounds with close connection to the weight tying scheme.
> > > >
> > > > 2. The paper does use multi-layer deep models in its experiments. I might have been too harsh in saying that it has weak relationships with deep learning. However, I do not see any details on how the weight tying can be applied to layers other than the first. The experimental section mentions all layers are EERLs, which could be misleading since a simple linear layer can also be an EERL.
> > > >
> > > > Based on these considerations, I will increase the score bit, but will still recommend rejection.

---

> > > > > ### Author Response · Authors · 2019-11-13
> > > > > **RE: multiple layers**
> > > > >
> > > > > Thank you for your update:
> > > > >
> > > > > REVIEW: "...However, I do not see any details on how weight tying can be applied to layers other than the first.  The experimental section mentions all layers are EERLs, which could be misleading since a simple linear layer can also be an EERL"
> > > > >
> > > > > RESPONSE: You may think of our layer analogous to a convolution layer for images: it uses parameter-sharing to achieve equivariance, and you may stack multiple layers, followed by nonlinearity. We prove that our proposed layer is the most general linear layer with this property, and it is the ONLY layer satisfying the condition of the definition of EERL (Equivariant Entity Relationship Layer); see Theorems 4.1 and 4.2.
> > > > >
> > > > > If there are any further issues we would be happy to respond and clarify.

---

### Official Review · AnonReviewer4 · 2019-10-31
**Official Blind Review #4**

**Rating:** 3

**Review:**

The authors extend recent work in equivariant set encoding to the setting of entity-relation data.  Similar to the cited previous work, they encode sets of objects (in this case the tuples of a relational database) with a permutation invariant function. They use a parameter tieing scheme to enforce this invariance.

I don’t find the paper to be particularly well motivated. Relational DBs are not necessarily a setting where you would always want equivariance. While the ordering of tuples do not matter, relations are often asymmetrical. Further, the idea of concatenating all tuples of a relational DB to be passed through a particular feed forward layer is infeasible for all but the smallest datasets. Real world databases such as knowledge bases contains millions to billions of entries. Scaling issues aside, the experiments do not actually show that this method outperforms any reasonable baselines such as a simple tensor factorization.

I also found it particularly hard to follow the descriptions of the methods. A few specific points:
- The text and notation in the beginning of section 2 could be a lot clearer. I had to read these paragraphs multiple times to pick out precisely what you were trying to say. Maybe have the set of relations be [R] like your other sets rather than a different script R.
- In the next paragraph you say “A particular relation R is a multiset if it contains multiple copies of the same entity” but you previously defined R to be a set of instances and not entities. I think you need to be more consistent with your terminology since the differences between type level entities and instances is important for your definitions and as you noted in your first footnote, these terms are often used very differently.
- This explanation could be helped a lot by improving the figure. The caption and images are both very dense but still requires a lot of coreference. For example, labeling the entities and relations with their ids in figure 1a directly would make it much easier to mentally map to what you are explaining in the caption. Also figure 1c is not at all clear.

Lastly, the methodology seems quite incremental over the previous work. A lot of the context and background that was sent to the appendix should be included in the main paper, particularly the relation to related work


edits:
double ‘the’ in abstract “linear complexity in the the data “

**Experience Assessment:**

I have read many papers in this area.

**Review Assessment: Checking Correctness Of Derivations And Theory:**

I assessed the sensibility of the derivations and theory.

**Review Assessment: Checking Correctness Of Experiments:**

I assessed the sensibility of the experiments.

**Review Assessment: Thoroughness In Paper Reading:**

I read the paper at least twice and used my best judgement in assessing the paper.

---

> ### Author Response · Authors · 2019-11-07
> **RE: Review #4**
>
> Thank you for your review.
>
> REVIEW: "I don’t find the paper to be particularly well motivated. Relational DBs are not necessarily a setting where you would always want equivariance. While the ordering of tuples do not matter, relations are often asymmetrical."
>
> RESPONSE: The notion of “asymmetry” of relations raised in the review is unclear to us. If for example, you mean “student-course” relation is different from “course-student” relation, then we agree. However, we do not understand your concern. Could you please elaborate?
>
> REVIEW: "Further, the idea of concatenating all tuples of a relational DB to be passed through a particular feed forward layer is infeasible for all but the smallest datasets. Real world databases such as knowledge bases contains millions to billions of entries.  "
>
> RESPONSE: Practicality is an important point that we also address in the paper. The model produced here does not perform any subsampling of the database, which is essential for its application to large datasets. We have seen successful examples of subsampling for graphs, and matrices recently. In the case of a relational database, subsampling is more involved, as one has to deal with variable amounts of sparsity across different relations (tables). This is left for future work.
>
> REVIEW: "Scaling issues aside, the experiments do not actually show that this method outperforms any reasonable baselines such as a simple tensor factorization."
>
> RESPONSE: We could not use tensor-factorization as a baseline in our experiments. Tensor factorization applies to settings where we have a single relation — for example, a user-move relation, or user-item-tag relation. This is the setting studied by Hartford et al’18. Since we show that our model reduces to their model in this setting, comparison to tensor factorization would amount to reproducing their results. To use the language of tensor factorization in our setup: we have multiple tensors that share some dimensions and need to be jointly factorized. We do not know of any factorization method for this task.
>
>
> REVIEW: "- The text and notation in the beginning of section 2..."
>
> RESPONSE: Based on your feedback we will try to make this part more clear (note that Using [R] for the set of relations would be misleading as the notation [X] is often used when X is ordinal or has an ordinal index, while in our paper, R is defined to be a set.)
>
> REVIEW: "In the next paragraph you say “A particular relation R is a multiset if it contains multiple copies of the same entity” but you previously defined R to be a set of instances and not entities. I think you need to be more consistent with your terminology..."
>
> RESPONSE: We never defined R as a set of instances (could you please identify lines that misled you? Since this is basic notation used throughout the paper, it is important we fix any ambiguity.)

---

> > ### Comment · AnonReviewer4 · 2019-11-13
> > **rebuttal response**
> >
> > After reading the other reviews and responses I slightly increased my overall rating of the paper.
> >
> > My biggest issues are still with the evaluations.
> > There are methods for embedding relational data with more than one relation. A non exhaustive review from a few years ago can be found here: https://persagen.com/files/misc/Wang2017Knowledge.pdf . Is there a reason these methods are not appropriate baselines, particularly for predicting missing records? Let me know if I'm misunderstanding or overlooking something.

---

> > > ### Author Response · Authors · 2019-11-13
> > > **knowledge-graph vs entity-relationship**
> > >
> > > Thank you for your follow-up.
> > >
> > > We understand that both knowledge graphs and the entity-relationship model are referred to as “relational” in the AI/ML community. While they have similarities, the models are not directly comparable; see Appendix A.1. for a detailed discussion.
> > >
> > > The suggested paper is for knowledge-graph embedding. A major difference is that knowledge-graphs do not have special accommodation for entity types. This difference also explains why deep models used for knowledge-graph completion are based on graph neural networks (e.g., Schlichtkrull et al’18). There are also graph databases (Robinson et al'13), that follow the semantics of knowledge graphs, but in this work, we are concerned with relational databases that use the entity-relationship diagrams.
> > >
> > > In addition to the appendix, we briefly address this connection to knowledge-graphs also in the introduction. If the reviewer finds it useful we are happy to add further discussion.

---

### Official Review · AnonReviewer5 · 2019-10-31
**Official Blind Review #5**

**Rating:** 3

**Review:**

This paper proposes Equivariant Entity-Relationship Networks, the class of parameter-sharing neural networks derived from the entity-relationship model.

Strengths of the paper:
1. The paper is well-written and well-structured.
2. Representative examples, e.g., the Entity-Relationship diagram in Figure 1, are used to demonstrate the proposed algorithms.
3. Detailed proofs for some equations are provided for better understanding the proposed equivariant entity-relationship networks.

Weaknesses of the papers:
1. No effective baselines are used for comparisons in the experiments. Are there state-of-the-art algorithms that have been proposed by other researchers to be used as baselines in the experiments?
2. No effective real-world datasets are used in the experiments. The authors only take synthesized toy dataset in their experiments. Are there other real-world datasets to be used in the experiments?
3. In terms of missing record prediction, why do the authors embed, e.g., the COURSE, in this way but not the other ways? What are the motivations of embedding like this? Are there other embedding techniques, e.g., Matrix Factorization and Skip-gram frameworks like that in Word2VEC, can be used for your purposes?


**Experience Assessment:**

I have read many papers in this area.

**Review Assessment: Checking Correctness Of Derivations And Theory:**

I assessed the sensibility of the derivations and theory.

**Review Assessment: Checking Correctness Of Experiments:**

I assessed the sensibility of the experiments.

**Review Assessment: Thoroughness In Paper Reading:**

I read the paper thoroughly.

---

> ### Author Response · Authors · 2019-11-07
> **RE: Review #5**
>
> Thank you for your review.
>
> REVIEW: "No effective baselines are used for comparisons in the experiments. Are there state-of-the-art algorithms that have been proposed by other researchers to be used as baselines in the experiments?"
>
> RESPONSE: We are unaware of any other methods applicable to the relational data in the form that appears in databases; we refer you to the detailed discussion of related work in Appendix A. We believe this should further count towards the novelty and timeliness of our contribution. Please let us know if you have any particular method in mind.
>
> REVIEW: "No effective real-world datasets are used in the experiments. The authors only take synthesized toy dataset in their experiments. Are there other real-world datasets to be used in the experiments?"
>
> RESPONSE: One of our experiments is on real-world data from a Kaggle (prediction of the outcomes in the European Soccer League). This dataset is at the limit where we can still pass the whole data through our model on a typical GPU (this can be verified by running our linked code). Note that due to lack of subsampling techniques, at this point we can only work with small datasets.
>
> REVIEW: "In terms of missing record prediction, why do the authors embed, e.g., the COURSE, in this way but not the other ways? What are the motivations of embedding like this? Are there other embedding techniques, e.g., Matrix Factorization and Skip-gram frameworks like that in Word2VEC, can be used for your purposes?"
>
> RESPONSE: We do not know of any embedding method that can be applied to relational data with more than one relation. Matrix and tensor factorization methods apply to settings where we have a single relation (across potentially multiple entities) — for example a STUDENT-COURSE relation, or STUDENT-COURSE-PROF relation. This is the setting studied by Hartford et al’18. Since we show that our model reduces to their model in this setting, comparison to the matrix and tensor factorization would amount to reproducing their results. In our setting, we have multiple tensors (see figure 1) that share some of their dimensions, as expressed by the ER diagram, and we need to perform joint factorization. We are not aware of any factorization methods for this setting. We will make this clear in the revised version.
>
> Could you please elaborate on what you mean by embedding the COURSE in this way?

---

### Official Review · AnonReviewer1 · 2019-11-01
**Official Blind Review #1**

**Rating:** 8

**Review:**

This paper defines a parameter-tying scheme for a general feed-forward network which respects the equivalence properties of a vector encoding of relational (ie database) data. As long as this parameter-tying is observed, the resultant feed-forward network is guaranteed to respect the equivariant properties of this data, meaning entity indexes can be permuted, and the output produced by the layer will be permuted in the correct manner. This results in a maximally expressive feed forward layer for this kind of data.

This is a very good paper and should be accepted. The approach is convincing, and provides a clear way forward for representing an important class of data. The paper is surprisingly well-written and comprehensible given the complexity of the math. I learned a lot from reading this paper.

The experiments are somewhat rudimentary, but they are perfectly adequate for a paper whose contribution is primarily methodological and theoretical.

One suggestion I have is that some of the details from the Appendix should be moved into the paper proper, most notably the details of the experiments in Appendix H.

**Experience Assessment:**

I have read many papers in this area.

**Review Assessment: Checking Correctness Of Derivations And Theory:**

I assessed the sensibility of the derivations and theory.

**Review Assessment: Checking Correctness Of Experiments:**

I assessed the sensibility of the experiments.

**Review Assessment: Thoroughness In Paper Reading:**

I read the paper thoroughly.

---

> ### Author Response · Authors · 2019-11-07
> **RE: review #1**
>
> Thanks for your review.
>
> We are happy to see your positive assessment. We will submit a revised version moving some of the experimental details to the main body (if you have a particular detail in mind please let us know.)

---

### Official Review · AnonReviewer3 · 2019-11-03
**Official Blind Review #3**

**Rating:** 3

**Review:**

This paper introduces a parameter tying scheme to express permutation equivariance in entity-relationship networks that are ubiquitous in relational databases. Results on the generality of the tying scheme are presented, as well as some experimental validation.

The paper extends several previously-proposed lines of work in statistical relational learning, including the work of Hartford et al. (2018) to multiple relations. In contrast to this earlier paper, which builds its argument step by step with great clarity (helped by illustrations), the current paper would prove hard to read to an ICLR audience, since most of its language and methods borrow more from the database theory literature. Since the proposed tying scheme is quite close to that of Hartford et al. (2018), one way to present the paper could be to construct the argument in such a way as to emphasize the difference with the previously-proposed scheme, thereby far better outlining the current contribution.

In addition, the experimental validation leaves much to be desired. In the synthetic data experiments (e.g. Figure 3), there is no guidance as to how to judge the quality of the resulting embedding. Also, it would be helpful to contrast the performance of the proposed approach against alternatives on a wider variety of datasets, such as MovieLens, Flixster, Netflix. (See the methodology followed by Hartford et al., 2018).

All in all, even though the proposed approach could usefully extend the state of the literature, the paper in its current form would need additional work before recommending acceptance at ICLR.


**Experience Assessment:**

I do not know much about this area.

**Review Assessment: Checking Correctness Of Derivations And Theory:**

I did not assess the derivations or theory.

**Review Assessment: Checking Correctness Of Experiments:**

I assessed the sensibility of the experiments.

**Review Assessment: Thoroughness In Paper Reading:**

I read the paper at least twice and used my best judgement in assessing the paper.

---

> ### Author Response · Authors · 2019-11-07
> **RE: Review #3**
>
> Thank you for your review.
>
> REVIEW: "The paper extends several previously-proposed lines of work  ... thereby far better outlining the current contribution"
>
> RESPONSE: We would like to clarify that we do not discuss any language, or theory beyond the “relational model” for databases, which is a minimal requirement. We use a running example along with several heavily captioned figures to help the reader get a visual summary of the main ideas. We would like to point out that the data that we work with, as well as our model and its analysis,  is inherently more complex than the model of Hartford et al’18.
>
> REVIEW:  "In the synthetic data experiments (e.g. Figure 3), there is no guidance as to how to judge the quality of the resulting embedding."
>
> RESPONSE: We agree that this could be made clearer in the paper, and will add a sentence to that effect. However, note that embedding plots are qualitative in nature. The figure demonstrates an agreement in ordering of the produced embedding and the ground truth (this is the best one can hope for).
>
> REVIEW: "also, it would be helpful to contrast the performance of the proposed approach against alternatives on a wider variety of datasets, such as MovieLens, Flixster, Netflix. (See the methodology followed by Hartford et al., 2018)."
>
> RESPONSE: We show that our model reduces to the model of Hartford et al’18 in the case of a single relation. The datasets proposed by the reviewer are such single-relation cases, and therefore running such experiments would simply be reproducing the results reported in Hartford et al.’18.

---

### Decision · Program_Chairs · 2019-12-19

**Decision:**

Reject

**Comment:**

This paper defines a parameter-tying scheme for a general feed-forward network with the equivalence properties of relational data. Most reviewers raised a few concerns around the experiments, baselines, datasets used and motivation. A few pointed out that the paper is hard to read - for a person without heavy database theory literature, which includes most of ICLR readers. While this paper may read well for the folks in the domain, authors should consider revising the paper to be more inclusive so that it can be read more widely. The motivation of the problem was also another point that many reviewers have mentioned (perhaps related to the language issues above) that some noted that you may not always want equivariance in relational DB and other noted that the paper did not sufficiently demonstrate the advantage of the proposed methods. Reviewers also univocally commented on experiments - many voiced the lack of baselines (not even any simple one). Authors wrote back to defend that there is no similar method and even simple tensor factorization isn’t applicable. That makes me wonder - is there really no single simple method you can compare with? If nobody had solution for this problem, is this a problem worth solving? Reviewers also encouraged to use larger (beyond Kaggle dataset) real-world datasets to strengthen the paper. All the points raised by reviewers suggests that this paper can benefit from another round of nontrivial editing before it’s ready for the show.